# The genetic profile of elite youth soccer players and its association with power and speed depends on maturity status

**Conall F. Murtagh**[1], **Thomas E. Brownlee**[1], **Edgardo Rienzi**[2], **Sebastian Roquero**[2], **Sacha Moreno**[2], **Gustavo Huertas**[2], **Giovani Lugioratto**[3], **Philipp Baumert**[1], **Daniel C. Turner**[1], **Dongsun Lee**[1], **Peter Dickinson**[1], **K. Amber Lyon**[1], **Bahare Sheikhsaraf**[1], **Betül Biyik**[1], **Andrew O'Boyle**[1,4], **Ryland Morgans**[5], **Andrew Massey**[6], **Barry Drust**[1], **Robert M. Erskine**[1,7] *

1 School of Sport and Exercise Sciences, Liverpool John Moores University, Liverpool, United Kingdom,
2 Club Atlético Peñarol, Estadio Campeón del Siglo, Montevideo, Uruguay, 3 Defensor Sporting Club, Montevideo, Uruguay, 4 Premier League, London, United Kingdom, 5 School of Sport and Health, Cardiff Metropolitan University, Cardiff, United Kingdom, 6 Liverpool Football Club, Liverpool, United Kingdom, 7 Institute of Sport, Exercise and Health (ISEH), University College London, London, United Kingdom

* R.M.Erskine@ljmu.ac.uk

**Data Availability Statement:** All relevant data are within the manuscript and its Supporting Information files.

## Abstract

We investigated the association of multiple single nucleotide polymorphisms (SNPs) with athlete status and power/speed performance in elite male youth soccer players (ESP) and control participants (CON) at different stages of maturity. ESP (n = 535; aged 8–23 years) and CON (n = 151; aged 9–26 years) were genotyped for 10 SNPs and grouped according to years from predicted peak-height-velocity (PHV), i.e. pre- or post-PHV, to determine maturity status. Participants performed bilateral vertical countermovement jumps, bilateral horizontal-forward countermovement jumps, 20m sprints and modified 505-agility tests. Compared to CON, pre-PHV ESP demonstrated a higher *ACTN3* (rs1815739) XX ('endurance') genotype frequency distribution, while post-PHV ESP revealed a higher frequency distribution of the *PPARA* (rs4253778) C-allele, *AGT* (rs699) GG genotype and *NOS3* (rs2070744) T-allele ('power' genotypes/alleles). *BDNF* (rs6265) CC, *COL5A1* (rs12722) CC and *NOS3* TT homozygotes sprinted quicker than A-allele carriers, CT heterozygotes and CC homozygotes, respectively. *COL2A1* (rs2070739) CC and *AMPD1* (rs17602729) GG homozygotes sprinted faster than their respective minor allele carrier counterparts in CON and pre-PHV ESP, respectively. *BDNF* CC homozygotes jumped further than T-allele carriers, while ESP *COL5A1* CC homozygotes jumped higher than TT homozygotes. To conclude, we have shown for the first time that pre- and post-PHV ESP have distinct genetic profiles, with pre-PHV ESP more suited for endurance, and post-PHV ESP for power and speed (the latter phenotypes being crucial attributes for post-PHV ESP). We have also demonstrated that power, acceleration and sprint performance were associated with five SNPs, both individually and in combination, possibly by influencing muscle size and neuromuscular activation.

**Funding:** We declare that seven co-authors are employed by commercial companies: Liverpool Football Club (AM), Club Atlético Peñarol (ER, SR, SM, GH), Defensor Sporting Club (GL) and the Premier League (AOB). However, these companies only provided support in the form of salaries for authors AM, ER, SR, GL, SM, GH and AOB, but did not have any additional role in the study design, data collection and analysis, decision to publish, or preparation of the manuscript. The specific roles of these authors are articulated in the 'author contributions' section.

**Competing interests:** The authors' affiliations with commercial companies do not represent any competing interests and do not affect the authors' roles stipulated in the 'author contributions' section of the manuscript. Further, this does not alter our adherence to PLOS ONE policies on sharing data and materials.

## Introduction

Human physical performance is influenced by a number of environmental and genetic factors [1]. The physical determinants of elite youth soccer playing status are specific to the stage of maturity [2], with power being more important in physically mature- *vs.* immature elite youth soccer players (ESP) [2]. However, it is not known if genetic variation can account for the differences in speed and power between ESP and recreationally active controls (CON) at different stages of maturity [2]. Such information could help inform genetic screening criteria in ESP, which could help practitioners identify talented pre-pubescent players, who are more likely to continue to succeed at an elite level when physically mature. Furthermore, talented pre-pubescent players found to have a less favourable genetic profile for post-pubescent physical performance could undergo personalised training to improve certain performance characteristics.

The majority of previous genetics research in soccer comprises case control studies in mainly senior players [3,4]. However, such studies are of limited use to the applied practitioner, as unlike cross-sectional genotype-phenotype association studies, they do not reveal the association between specific genetic polymorphisms and physical performance phenotypes. There is a paucity of cross-sectional genotype-phenotype studies in soccer [5–7] and, importantly, the contribution of genetic variation to power, speed and agility performance in ESP remains unknown.

Over 69 genetic markers have been associated with power athlete status [8] and the genetic association with power and speed is likely due to specific gene variants influencing protein abundance/expression, which will in turn affect the physiological determinants of speed and power. Considering this, patellar tendon compliance is a key physiological determinant of horizontal-forward jump performance in under 18 year-old (U18) and U21 ESP [9], and has previously been associated with sprint performance [10]. The *COL5A1* (rs12722) single nucleotide polymorphism (SNP) has been associated with the extensibility of the tendon-aponeurosis structures of the knee extensors in some [11], but not all [12] studies. Other collagen SNPs, such as *COL1A1* (rs2249492) and *COL2A1* (rs2070739), which are variants of the genes encoding the alpha 1 chain of procollagen type I and II, respectively, may also influence tendon properties and therefore power performance, but this has yet to be investigated.

In contrast to horizontal-forward jump performance, one of the main physiological determinants of vertical jump performance in soccer players is quadriceps femoris muscle volume [13]. The *NOS3* (rs2070744) T [14] and *AGT* (rs699) G [15,16] alleles have both been associated with elite power athlete status and are thought to exert their favourable effect on power performance by promoting skeletal muscle hypertrophy [14,16]. However, these SNPs have not been investigated in association with power/speed performance in ESP.

In addition to muscle volume, muscle fibre-type composition can also influence peak power output, with type II fibres being larger (therefore producing more force) and able to shorten quicker, thus able to generate more power, than type I fibres [17]. The *ACTN3* (rs1815739) R-allele and *PPARA* (rs4253778) C-allele have previously been associated with ESP status [3,4] and a greater composition of type II skeletal muscle fibres [18] but evidence linking the *ACTN3* SNP with power/speed in professional soccer players is equivocal [5,6]. Therefore, it is still unclear whether these SNPs influence power/speed performance in ESP at different stages of maturity.

Maximal power during sprinting and jumping is not only governed by muscle-tendon properties, but also neuromuscular activation [13,19]. Brain derived neurotrophic factor (BDNF) is a neurotrophin that regulates neuronal survival, growth, maintenance, neurogenesis and synaptic plasticity, and the *BDNF* (rs6265) C>T SNP has been associated with BDNF serum concentration in response to exercise [20]. It is possible that this SNP might influence

neural adaptation and, therefore, the ability to activate the relevant muscles and generate more power during jump and sprint performance. However, this has yet to be investigated in ESP.

Furthermore, the *AMPD1* (rs17602729) GG genotype has been associated with elite power athlete status [21] but has never been investigated in ESP. The *VDR* (rs2228570) GG genotype, on the other hand, has been associated with average- to high-level youth soccer playing status but was not associated with vertical jump, 10 m acceleration or 20 m sprint performance in adolescent soccer players [7]. However, this study was limited by a relatively low sample size (n = 125) and did not account for the confounding factor of maturity status. It therefore remains unknown if the *VDR* (rs2228570) A>G SNP is associated with power/sprint performance in ESP at different stages of maturity.

We aimed to investigate the association of 10 candidate SNPs with ESP playing status at different stages of maturity, and the individual and combined association of those SNPs with power/speed performance in ESP and CON according to maturity status. We hypothesised that the physically mature players would have a more power-orientated genetic profile than the younger players, and that individuals possessing multiple 'power' genotypes would perform better in the power/speed assessments within their respective maturity groups.

## Materials and methods

All procedures performed in this study involving human participants were approved by Liverpool John Moores University Research Ethics Committee (protocol number: P13/SPS/033) and were in accordance with the 1964 Helsinki Declaration and its later amendments. Written informed consent was obtained from all individual study participants (and parents/guardians if participants <18 years old).

### Participants

Six hundred and eighty-six males volunteered to take part in this study and formed two cohorts: 535 ESP and 151 CON. The ESP (U9 to U23 level) were considered elite if they had competed regularly (played over 12 competitive games in the previous 6 months) for a Category One (highest national category) soccer academy in England, or one of two Category A (highest national category) soccer academies in Uruguay. The age-matched CON were young, healthy males, who had not previously played soccer at academy or professional level. All participants were free from any lower-limb injury to the lower body in the three months prior to the study and had not previously sustained a serious knee or ankle injury that may affect performance during the study. Ethnic background was reported for all ESP (white, 79%; black, 5%; black/white, 16%; and Asian, 1%) and a sub-sample (*n* = 58) of CON (white, 93%; black, 2%; mixed black/white, 3%; and Asian, 2%). Participant characteristics are displayed in Table 1.

### Experimental design

Allele/genotype frequency distribution for 10 SNPs was compared between all ESP and all CON at different stages of maturity. A list of the SNPs, their associated genes, proteins and functions are presented in Table 2. Of the 686 participants, 303–602 performed power/speed assessments (depending on the assessment, exact sample sizes are stipulated in the relevant sections, below). Power assessments included a bilateral horizontal countermovement jump (BH CMJ) and a bilateral vertical countermovement jump (BV CMJ). Speed assessments included acceleration (10 m straight line sprint), sprint (20 m straight line sprint) and agility (modified 505 agility test) performance tests. We investigated associations between the 10 different SNPs and power/speed performance in ESP and CON, according to maturity status. In

**Table 1. Participant characteristics for the elite soccer player (ESP) and control (CON) cohorts, according to maturity status, i.e. pre-, mid- or post-peak height velocity (PHV).**

| Cohort | Maturity group ($n$) | Age (years) | Height (m) | Body mass (kg) |
|--------|----------------------|-------------|------------|----------------|
| ESP    | Pre-PHV (124)        | 10.6 ± 1.4  | 1.44 ± 0.09 | 35.7 ± 6.0    |
|        | Mid-PHV (43)         | 14.1 ± 0.8  | 1.64 ± 0.06 | 51.8 ± 7.5    |
|        | Post-PHV (368)       | 16.8 ± 2.3  | 1.75 ± 0.12 | 69.1 ± 10.7   |
| CON    | Pre-PHV (44)         | 11.2 ± 1.3  | 1.45 ± 0.08 | 37.5 ± 5.8    |
|        | Mid-PHV (30)         | 13.9 ± 0.5  | 1.62 ± 0.06 | 52.0 ± 7.3    |
|        | Post-PHV (77)        | 17.5 ± 3.2  | 1.75 ± 0.06 | 68.7 ± 9.4    |

a sub-sample of 42 post-PHV participants (ESP, $n = 22$; CON, $n = 20$), quadriceps femoris (QF) muscle physiological characteristics were determined. These included knee extension maximum voluntary contraction (MVC) moment, muscle volume, muscle physiological cross-sectional area (PCSA) and vastus lateralis muscle fascicle pennation angle. All tests were performed during the in-season period and testing sessions were scheduled >48 h after competition or a high intensity training session to minimize the influence of prior exercise.

## Anthropometry

Height, sitting height and body mass were assessed as previously described [2]. Pubertal timing was calculated according to the estimated biological age of each individual, using calculations described elsewhere [40]. The age at which peak linear growth in stature occurs [i.e. peak height velocity (PHV)] is an indicator of somatic maturity, and biological age was calculated by subtracting the chronological age at the time of testing from the estimated chronological age at PHV. Participants were segregated into two maturity groups based on biological age, i.e. pre-PHV ($\leq$ -1.0 years), mid-PHV (-0.99 to 0.5 years) and post-PHV (> 0.5 years). The number of participants in each maturity group is presented in Table 1.

## Warm up protocol, jump and speed assessments

The warm up, jump and speed assessments were performed as detailed elsewhere [2] and are described briefly, below. Participants wore a t-shirt, shorts and football boots for all tests except the BV CMJ, for which no footwear was permitted.

**Warm up protocol.** The standardized 10 min warm up comprised 5 min dynamic movements (e.g. high knees, skips, lunges), followed by at least 3 practice trials at each assessment (after a demonstration of each).

**Jump assessments.** Participants performed a minimum of 3 trials of the BH CMJ (ESP $n = 471$; CON $n = 131$) and BV CMJ (ESP $n = 211$; CON $n = 92$) with approximately 30 seconds of recovery between trials and 5 min between jump types. If the third jump measurement (height or distance) was higher than the first or second, the participant performed a fourth trial. The highest or longest jump was selected for subsequent analysis.

**Acceleration/Sprint assessments.** A photocell timing system (Brower Timing System, Salt Lake City, UT, USA) was used to assess sprints to the nearest 0.001 s. Participants (ESP $n = 458$; CON $n = 111$) were required to perform three maximal sprints in which they were instructed to run 24 m as quickly as possible. The first, second and third timing gates were positioned 1 m, 11 m and 21 m from the start line, respectively. Thus, acceleration was defined as the time taken to run 0–10 m, and sprint was defined as the time taken to run 0–20 m.

**Agility (change of direction) assessment.** A timing gate (Brower) was positioned 1 m in front of the start line. After assuming a split stance crouch position, with their front foot

**Table 2. Information pertaining to the 10 single nucleotide polymorphisms (SNPs) investigated in the current study.** The location of each SNP is identified according to Genome Reference Consortium Human Build 38 patch release 12 (GRCh38.p12) and alleles for each SNP are reported according to the forward DNA strand.

| Gene | Encoded protein | Protein function | Alleles and rs number | Location | MAF | SNP Type | SNP function |
|------|-----------------|------------------|-----------------------|----------|-----|----------|--------------|
| ACTN3 | α-actinin-3 | Binds actin to the Z-line in type II SkM fibres [22]; blocks calcineurin, inhibiting slow myogenic programme [23] | C>T rs1815739 | chr11:66560624 | T: 0.434 | Transition substitution, nonsense mutation, intragenic | T-allele leads to a stop codon at amino acid 577, preventing protein production [24] |
| AGT | Angiotensinogen | Precursor of angiotensin II, which modulates SkM hypertrophy in response to mechanical loading [25] | A>G rs699 | chr1:230710048 | G: 0.412 | Transition substitution, missense mutation, intragenic | G-allele associated with higher plasma AGT levels and higher blood pressure [26] |
| AMPD1 | Adenosine Monophosphate Deaminase 1 | Enzyme catalysing the deamination of AMP to IMP in SkM | G>A rs17602729 | chr1:114693436 | A: 0.123 | Missense mutation, intron, transition substitution, intragenic | A-allele results in a premature stop codon and non-functional enzyme, causing impaired AMP metabolism, which can lead to muscle fatigue, weakness and cramping [27] |
| BDNF | Brain-derived neurotrophic factor | Promotes neurone growth, differentiation, and maintenance [28] | C>T rs6265 | chr11:27658369 | T: 0.197 | Transition substitution, missense mutation, intragenic | C-allele results in valine rather than methionine (T-allele) at amino acid 66, leading to greater abundance of exercise-induced serum BDNF concentration [20] |
| COL1A1 | Pro-α1 (I) chain | Major component of type I collagen, a structural protein found in most connective tissues, including tendon [29] | C>T rs2249492 | chr17:50185660 | C: 0.379 | Transition substitution, intron, intragenic | Not yet known but intronic SNPs have the potential to influence gene expression and mRNA stability [30] |
| COL2A1 | Pro-α1 (II) chain | Major component of type II collagen, providing structure and strength to connective tissue, e.g. at the enthesis [31] | C>T rs2070739 | chr12:47974193 | T: 0.096 | Transition substitution, missense mutation, intragenic | Not yet known |
| COL5A1 | Pro-α1 (V) chain | Major component of type V collagen, which regulates the diameter of collagen fibrils [32] | C>T rs12722 | chr9:134842570 | C: 0.415 | Transition substitution, 3'-UTR, intragenic | Not yet known but 3'-UTR SNPs have the potential to alter the level, location, or timing of gene expression [30] |
| NOS3 | Nitric oxide synthase-3 | Potentially stimulating muscle hypertrophy through NO-mediated vasodilatation[14] | C>T rs2070744 | chr7:150992991 | C: 0.438 | Transition substitution, intron, intragenic | T-allele increases gene promoter activity, thus increasing eNOS and NO synthesis [33] |
| PPARA | Peroxisome proliferator-activated receptor-α | Present in SkM and promotes uptake, utilization, and catabolism of fatty acids (FAs) by upregulation of genes involved in FA transport, FA binding activation, and peroxisomal and mitochondrial FA β-oxidation [34] | G>C rs4253778 | chr22:46234737 | C: 0.192 | Intron, transversion substitution, intragenic | Not yet known but intronic SNPs have the potential to influence gene expression and mRNA stability [30] |
| VDR | Vitamin D receptor | Downstream targets of VDR are involved in SkM regeneration, recovery and hypertrophy following strenuous exercise [35] | A>G rs2228570 | chr12:47879112 | A: 0.378 | Missense mutation, transition substitution, intragenic | G-allele leads to 244 amino acid-long VDR, while A-allele produces 247 amino acid-long protein. The shorter VDR has enhanced transactivation capacity as a transcription factor [36–38] |

SNP, single nucleotide polymorphism; MAF, minor allele frequency (according to European population [39]); chr, chromosome; SkM, skeletal muscle.

behind the start line, participants (ESP $n$ = 439; CON $n$ = 38) were instructed to sprint to a line 5 m further from the start line, plant their right foot on this line, turn and sprint back in a straight line past the original start line. The time taken for the participants to run the full 5 m plus 5 m distance, i.e. breaking the infrared beam of the timing gate twice, was recorded using a hand-held wireless controller. This procedure was performed three times using the right leg, and three times using the left leg as a pivot. The fastest time of the three sprints was recorded and represented agility using that particular leg as a pivot. Participants performed this "*modified 505 agility test*" on a 4G artificial grass surface.

**Quadriceps femoris muscle physiological characteristics.** The protocols for measuring QF muscle strength, volume, PCSA and architecture has been described in detail previously [41]. A summary of these procedures is outlined below.

*Muscle strength*. Knee extension isometric MVCs of the dominant leg were assessed on an isokinetic dynamometer (Biodex 3, Medical Systems, Shirley, USA) and analysed using Acq-Knowledge data acquisition software (Biopac Systems Inc., Goleta, CA, USA). Participants sat on the rigid chair with their hip angle set to 85˚ (supine position was equivalent to 180˚) and their knee angle set to 90˚ (0˚ = full extension), before being secured at the hip, chest and distal thigh with inextensible straps to minimise movement. Participants then performed at least three MVCs lasting 2–3 s, with 60 s rest in between attempts. If the highest of the three MVC swas >5% higher than the next highest, more attempts were made to ensure true MVC was achieved.

*Muscle volume ($V_m$)*. With the participant in a relaxed seated position on the isokinetic dynamometer (knee joint angle at 90˚), B-mode ultrasonography (MyLab 30 CV, Esoate Bio-medica, Genoa, Italy) was used to locate the distal (lateral femoral condyle) and proximal (base of greater trochanter) ends of the femur, with the distance between both points providing the femur length. The anatomical cross-sectional area (ACSA) of the quadriceps was then measured at 40% of femur length (from the distal end) using ultrasound [42]. $V_m$ was calculated from a combination of this ACSA, femur length, and a series of regression equations detailed elsewhere [43].

*Vastus lateralis (VL) muscle fascicle length ($L_f$) and pennation angle ($\theta_p$)*. VL $L_f$ and $\theta_p$ were measured at rest using ultrasonography with the participant in a relaxed seated position on the isokinetic dynamometer (knee joint angle at 90˚). Once the origin and insertion of the vastus lateralis were identified, this enabled the lateral and medial boundaries of the muscle to be located at 50% of its length. The centre of the muscle was then identified and marked on the skin with a permanent marker pen, and this location was used to measure $L_f$ and $\theta_p$ in three fascicles, and the mean of those three was used for subsequent analysis.

*Physiological cross sectional area (PCSA)*. QF muscle PCSA was calculated by dividing QF $V_m$ by VL $L_f$ [40].

**Blood and saliva sampling.** Participants' DNA was obtained from either whole blood (n = 83) or saliva (n = 603) samples. For blood sampling, 10-mL was drawn from a superficial forearm vein into an EDTA vacutainer (BD Vacutainer Systems, Plymouth, UK). The whole blood was then aliquoted into 2-mL cryotubes (Eppendorf AG, Hamburg, Germany) and stored at -80˚C until DNA extraction. For saliva samples, the participants dribbled 2 mL saliva into a Genefix saliva collection tube (Isohelix Ltd., Harrietsham, UK), having not eaten or drunk anything (or chewed gum) for at least 30 min prior to saliva sampling. After collection, the saliva tubes were closed and gently shaken to mix the saliva with 2 mL non-toxic stabilization buffer contained within the tube. Samples were then aliquoted into 2 mL cryotubes (Eppendorf AG) and stored at -80˚C until DNA extraction.

**DNA extraction and genotyping.** DNA purification from whole blood and saliva samples was performed manually using a QIAamp® DNA Blood Mini Kit (Qiagen Ltd., Manchester,

UK), following the manufacturer's guidelines. DNA samples were then stored at 4˚C until subsequent genotyping.

Real-time polymerase chain reaction was performed (Rotor-Gene Q, Qiagen) to establish the genotypes of each SNP for each participant. Each 10 $\mu$L reaction volume contained 5 $\mu$L Genotyping Master Mix (Applied Biosystems, Foster City, USA), 3.5 $\mu$L nuclease-free $H_2O$ (Qiagen), 0.5 $\mu$L genotyping assay (Applied Biosystems), plus 1 $\mu$L DNA sample. Both negative [1 $\mu$L nuclease-free $H_2O$ (Qiagen) replaced the DNA template] and positive controls were included in each RT-PCR run, which used the following protocol: denaturation at 95˚C for 10 min, followed by 50 cycles of incubation at 92˚C for 15 s, then annealing and extension at 60˚C for 1 min. Genotypes were determined using Rotor-Gene Q Pure Detection 2.1.0 software (Qiagen). All samples were analysed in duplicate and there was 100% agreement between genotype calls for samples from the same participant. Genotyping was performed in accordance with published genotyping and quality control recommendations [44] and all SNPs were genotyped according to the forward DNA strand.

## Statistical analysis

Genotype frequency distributions for all ESP, pre-PHV ESP, post-PHV ESP and CON were tested for compatibility with Hardy-Weinberg equilibrium (HWE) using $\chi^2$ goodness of fit tests. Allele and genotype frequency distribution for each SNP were compared between all ESP and CON, and between pre- and post-PHV ESP and CON, using $\chi^2$ tests of homogeneity (Table 3). The mid-PHV ESP group was omitted from the latter analyses due to its relatively small sample size. To control for multiple SNP comparisons, a false discovery rate (FDR) of 0.2 [45] was applied to both the allele and genotype frequency distribution models. For both allele and genotype, two FDR models (each including 10 *P*-values, one for each SNP) were calculated for each population comparison, i.e. (i) all ESP *vs*. all CON; and (ii) pre-PHV ESP *vs*. post-PHV ESP *vs*. all CON. The original *P*-values in each model were sorted separately in ascending order, and then each original *P*-value was compared with its respective Benjamini-Hochberg (B-H) critical value [45]. For each FDR model, the largest original *P*-value that was less than its B-H critical value was deemed significant, together with any other original *P*-values smaller than that [45].

Those SNPs that demonstrated differences were combined to create a total genotype score (TGS) [46], with the 'power/speed' homozygote genotype for each SNP given a score of 2, the heterozygote a score of 1, and the 'non-power/speed' homozygote a score of 0. This 'ESP' TGS was then compared between CON, pre-PHV ESP and post-PHV ESP with a one-way ANOVA.

For each of the 10 SNPs, a three-way between-subjects ANOVA was used to investigate the association between genotype (either 3 or 2 genotype groups, depending on the minor allele frequency, see Tables 2 and 3), player status (ESP *vs*. CON), maturity status (pre- *vs*. mid- *vs*. post-PHV) and physical performance (horizontal/vertical power, acceleration, sprint and agility). Thus, we focussed on the three-way interaction (genotype x athlete status x maturity status), two-way interactions (genotype x athlete status and genotype x maturity status), and main effect of genotype. To control for multiple comparisons (10 SNPs), an FDR of 0.2 [45] was applied to the SNP main effect and interaction effect *P*-values for each performance variable. Thus, four FDR models (each including 10 *P*-values, one for each SNP) were calculated for each phenotype, i.e. (i) the main effect of SNP; (ii) SNP x athlete status two-way interaction; (iii) SNP x maturity status two-way interaction; (iv) SNP x athlete status x maturity status three-way interaction. As above, *P*-values in each model were sorted separately in ascending order, and compared with their respective B-H critical values [45]. If a significant three-way

**Table 3. Allele and genotype frequency distributions for the 10 investigated single nucleotide polymorphisms (SNPs) in all elite youth soccer players (ESP) and control participants (CON), and in pre- and post-peak height velocity (PHV) ESP.** All SNPs were analysed using a co-dominant model (e.g. AA *vs.* Aa *vs.* aa) except when the minor allele frequency (MAF) for a European population [39] was <0.25, in which case a dominant model (e.g. AA *vs* Aa+aa) was used instead. *P*-values are presented for allele and genotype frequency distribution $\chi^2$ models (model 1: CON *vs.* ESP; and model 2: CON *vs.* pre-PHV ESP *vs.* post-PHV ESP) and post-hoc between group differences are identified (*P* < 0.05), i.e. * difference between ESP and CON; ** difference between post-PHV ESP and CON; *** difference between pre-PHV ESP and CON; † difference between post- and pre-PHV ESP.

| Cohort | | ESP | | | CON | $\chi^2$ model 1 | $\chi^2$ model 2 |
|---|---|---|---|---|---|---|---|
| Sub groups | | All | Pre-PHV | Post-PHV | All | CON *vs.* ESP | CON *vs.* pre-PHV ESP *vs.* post-PHV ESP |
| *n* | | 535 | 124 | 372 | 151 | *P*-values | *P*-values |
| *PPARA* G>C (rs4253778) | | | | | | | |
| Allele (%) | C | 24.1 | 19.0 | 25.8† | 18.5 | 0.042 | 0.011 |
| | G | 75.9 | 81.0 | 74.2† | 81.5 | | |
| Genotype (%) | CC+GC | 42.2* | 34.7 | 44.9**† | 31.8 | 0.021 | 0.009 |
| | GG | 57.8* | 65.3 | 55.1**† | 68.2 | | |
| *NOS3* C>T (rs2070744) | | | | | | | |
| Allele (%) | C | 35.3* | 38.7 | 34.8 | 42.7 | 0.019 | 0.057 |
| | T | 64.7* | 61.3 | 65.2 | 57.3 | | |
| Genotype (%) | CC | 13.3 | 16.1 | 12.4 | 18.5 | 0.069 | 0.218 |
| | CT | 44.1 | 45.2 | 45.2 | 48.3 | | |
| | TT | 42.6 | 38.7 | 42.5 | 33.1 | | |
| *ACTN3* C>T (rs1815739) | | | | | | | |
| Allele (%) | C (R) | 56.7 | 52.0 | 58.3 | 57.9 | 0.706 | 0.206 |
| | T (X) | 43.3 | 48.0 | 41.7 | 42.1 | | |
| Genotype (%) | CC (RR) | 35.7 | 33.1 | 36.6 | 31.8 | 0.061 | 0.038 |
| | CT (RX) | 42.1 | 37.9*** | 43.5 | 52.3 | | |
| | TT (XX) | 22.2 | 29.0*** | 19.9 | 15.9 | | |
| *AGT* A>G (rs699) | | | | | | | |
| Allele (%) | G | 50.5* | 44.8 | 53.5**† | 42.7 | 0.017 | 0.002 |
| | A | 49.5* | 55.2 | 46.5**† | 57.3 | | |
| Genotype (%) | GG | 25.0 | 21.8 | 28.0** | 16.6 | 0.051 | 0.008 |
| | GA | 50.8 | 46.0 | 51.1 | 52.3 | | |
| | AA | 24.1 | 32.3 | 21.0**† | 31.1 | | |
| *AMPD1* G>A (rs17602729) | | | | | | | |
| Allele (%) | G | 90.0 | 89.9 | 90.6 | 90.1 | 0.973 | 0.937 |
| | A | 10.0 | 10.1 | 9.4 | 9.9 | | |
| Genotype (%) | GG | 81.5 | 81.5 | 82.8 | 82.1 | 0.861 | 0.939 |
| | GA+AA | 18.5 | 18.5 | 17.2 | 17.9 | | |
| *BDNF* C>T (rs6265) | | | | | | | |
| Allele (%) | C | 83.2 | 79.8 | 84.5 | 82.1 | 0.666 | 0.204 |
| | T | 16.8 | 20.2 | 15.5 | 17.9 | | |
| Genotype (%) | CC | 69.0 | 63.7 | 71.2 | 66.9 | 0.626 | 0.249 |
| | CT+TT | 31.0 | 36.3 | 28.8 | 33.1 | | |
| *COL1A1* C>T (rs2249492) | | | | | | | |
| Allele (%) | T | 58.6 | 60.1 | 58.3 | 59.3 | 0.834 | 0.879 |
| | C | 41.4 | 39.9 | 41.7 | 40.7 | | |
| Genotype (%) | TT | 34.8 | 37.1 | 34.9 | 37.1 | 0.773 | 0.978 |
| | TC | 47.7 | 46.0 | 46.8 | 44.4 | | |
| | CC | 17.6 | 16.9 | 18.3 | 18.5 | | |
| *COL2A1* C>T (rs2070739) | | | | | | | |
| Allele (%) | C | 89.4 | 89.5 | 89.2 | 89.7 | 0.882 | 0.972 |

*(Continued)*

**Table 3.** (Continued)

| Cohort | | ESP | | | CON | χ² model 1 | χ² model 2 |
|---|---|---|---|---|---|---|---|
| Sub groups | | All | Pre-PHV | Post-PHV | All | CON *vs*. ESP | CON *vs*. pre-PHV ESP *vs*. post-PHV ESP |
| n | | 535 | 124 | 372 | 151 | *P*-values | *P*-values |
| | T | 10.6 | 10.5 | 10.8 | 10.3 | | |
| Genotype (%) | CC | 80.0 | 79.8 | 79.6 | 80.1 | 0.971 | 0.989 |
| | CT+TT | 20.0 | 20.2 | 20.4 | 19.9 | | |
| *COL5A1* C>T (rs12722) | | | | | | | |
| Allele (%) | C | 49.6 | 51.6 | 49.2 | 43.7 | 0.069 | 0.144 |
| | T | 50.4 | 48.4 | 50.8 | 56.3 | | |
| Genotype (%) | CC | 23.0 | 27.4 | 22.6 | 21.2 | 0.043 | 0.144 |
| | CT | 53.3 | 48.4 | 53.2 | 45.0 | | |
| | TT | 23.7 | 24.2 | 24.2 | 33.8 | | |
| *VDR* A>G (rs2228570) | | | | | | | |
| Allele (%) | G | 64.5 | 64.9 | 63.8 | 68.5 | 0.191 | 0.351 |
| | A | 35.5 | 35.1 | 36.2 | 31.5 | | |
| Genotype (%) | GG | 40.7 | 40.3 | 40.1 | 45.7 | 0.400 | 0.650 |
| | GA | 47.5 | 49.2 | 47.6 | 45.7 | | |
| | AA | 11.8 | 10.5 | 12.4 | 8.6 | | |

interaction occurred, simple simple main effects and simple simple pairwise comparisons with Bonferroni adjustment revealed significant differences. If a significant two-way interaction occurred, simple main effects and pairwise comparisons with Bonferroni adjustment were performed to reveal differences. If a significant genotype main effect existed, Bonferroni post-hoc tests were used to locate the genotype difference in test performance scores. We accounted for the larger ESP *vs*. CON cohort size by ensuring that all data were checked for violations of specific three-way ANOVA assumptions that may invalidate the interpretation of results. More specifically, Mauchly's test of sphericity, which assesses whether the variance of the differences between levels of the within-subject factors are equal, was applied to establish if the homogeneity of variance principle had been violated. If Mauchly's test of sphericity was violated, the Greenhouse-Geisser correction method was applied [47]. Box plots were also used to check for outliers (none were found); and all data sets were normally distributed according to the Kolmogorov-Smirnov test.

A separate 'performance' TGS [46] to the one described above was used to estimate the combined influence of *only* those SNPs that were individually associated with power/speed performance after FDR correction. This was based on a similar approach used by our group to investigate the polygenic association with exercise-induced muscle damage [48]. Spearman correlations were used to determine relationships between this TGS and speed/power performance according to PHV group.

Genotype differences in quadriceps femoris muscle strength, size and architecture were assessed via one-way ANOVAs. To control for multiple comparisons (10 SNPs and four physiological variables), an FDR of 0.2 [45] was applied to the SNP main effect *P*-values for each variable. As above, *P*-values in each model were sorted separately in ascending order, and compared with their respective B-H critical values [45]. If a significant main effect existed, Bonferroni pairwise comparisons were used to locate differences between genotypes.

Statistical analyses were completed using IBM SPSS Statistics 26 (IBM, New York, USA), and the significance level was set at $P < 0.05$.

## Results

### Allele and genotype frequency distributions

**Hardy-Weinberg equilibrium (HWE).**   Genotype frequency distributions of all 10 SNPs were in HWE in all CON and ESP except for the ACTN3 SNP, which was not in HWE in all ESP ($\chi^2$ = 11.430, P = 0.003). When ESP were segregated according to maturity status, pre- and post-PHV genotype distributions for all 10 SNPs were in HWE except for the ACTN3 SNP, which was not in HWE in pre-PHV elite soccer players ($\chi^2$ = 6.267, P = 0.044).

**Genotype and allele frequency distributions in ESP vs. CON.**   After FDR correction, four SNPs [*PPARA* G>C (rs4253778), *NOS3* C>T (rs2070744), *ACTN3* C>T (rs1815739), and *AGT* A>G (rs699)] demonstrated differences in allele/genotype frequency distribution between all ESP and CON, pre-PHV ESP and CON, post-PHV ESP and CON, or pre-PHV ESP and post-PHV ESP (Table 3).

*'ESP' TGS based on differences in allele/genotype frequency distributions between pre- and post-PHV ESP, and between pre-/post-PHV ESP and CON.* The 'ESP' TGS differed between groups (F = 9.064; *P*<0.001), with post-PHV ESP (50.7 ± 18.2) being higher than pre-PHV ESP (44.1 ± 19.0; *P* = 0.001) and CON (44.5 ± 18.5; *P* = 0.001), but no difference between pre-PHV ESP and CON (*P* = 0.848). Regarding muscle physiological variables in post-PHV ESP and CON (*n* = 42), the 'ESP' TGS correlated with quadriceps femoris (QF) muscle volume (*r* = 0.400; *P* = 0.010), QF PCSA (*r* = 0.334; *P* = 0.033), VL $\theta_p$ (*r* = 0.416; *P* = 0.007) and knee extension MVC (*r* = 0.408; *P* = 0.008). In post-PHV ESP only (*n* = 22), the 'ESP' TGS correlated with QF muscle volume (*r* = 0.430; *P* = 0.046) and QF PCSA (*r* = 0.438; *P* = 0.041), vastus lateralis $\theta_p$ (*r* = 0.441; *P* = 0.040) and knee extension MVC (*r* = 0.474; *P* = 0.026).

### Associations between genotype and power/speed performance

**Bilateral horizontal jump performance.**   There was a main effect of *BDNF* genotype (F = 7.367, *P* = 0.007; Fig 1A), with CC homozygotes achieving greater bilateral horizontal jump distance than T-allele carriers (regardless of athlete or maturity status).

**Bilateral vertical countermovement jump.**   Prior to FDR correction, there was a main effect of *AGT* genotype (F = 3.624, *P* = 0.028), with GG genotype jumping higher than both AG (*P*<0.001) and AA (*P*<0.001) genotypes but no difference between AG and AA (*P* = 0.150). However, following FDR correction, this main effect was no longer significant. There was a two-way interaction between *COL5A1* genotype and athlete status (F = 5.129, *P* = 0.004; Fig 3C), with ESP CC homozygotes achieving greater jump height than TT homozygotes (*P* = 0.002). There was also a non-significant tendency for CC homozygotes to achieve greater jump height than CT genotypes (*P* = 0.090).

**10 m acceleration performance.**   There was a main effect of *BDNF* genotype (F = 6.697, *P* = 0.010; Fig 1B), with CC homozygotes achieving faster 10 m acceleration times than T-allele carriers. There was a main effect of *COL2A1* genotype (F = 5.365, *P* = 0.021; Fig 3A), with CC homozygotes achieving faster 10 m acceleration times than T allele carriers. There was also a two-way interaction between *COL2A1* genotype and athlete status (F = 12.346, *P*<0.0001; Fig 3A), with CON CC homozygotes achieving faster 10 m acceleration times than T-allele carriers (*P* < 0.0001). There was a main effect of *COL5A1* genotype (F = 3.532, *P* = 0.030; Fig 2A), with CC homozygotes achieving faster 10 m acceleration times than CT genotype (*P* = 0.025). There was a main effect of *NOS3* genotype (F = 4.308, *P* = 0.014; Fig 4), with TT homozygotes achieving faster 10 m acceleration times than CC homozygotes (*P* = 0.011). There was a three-way interaction between *AMPD1* genotype, athlete status and maturity status (F = 4.061;

**A**

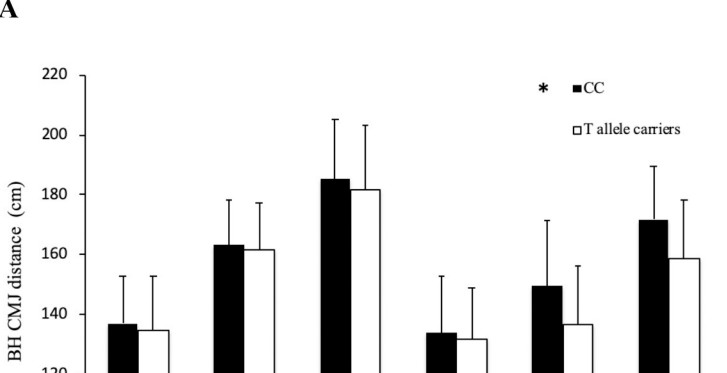

**B**

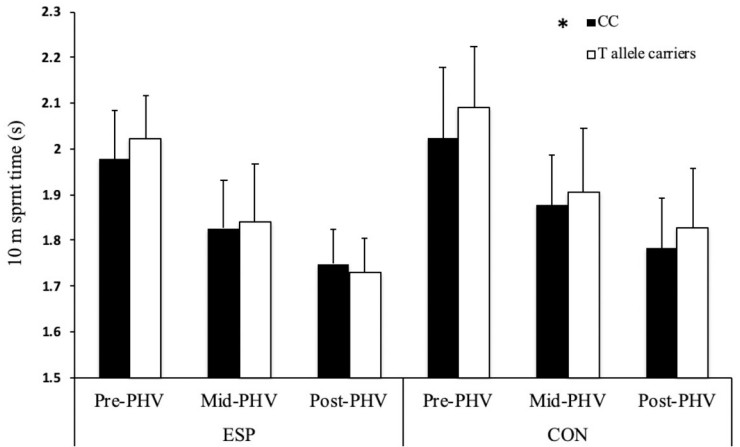

**C**

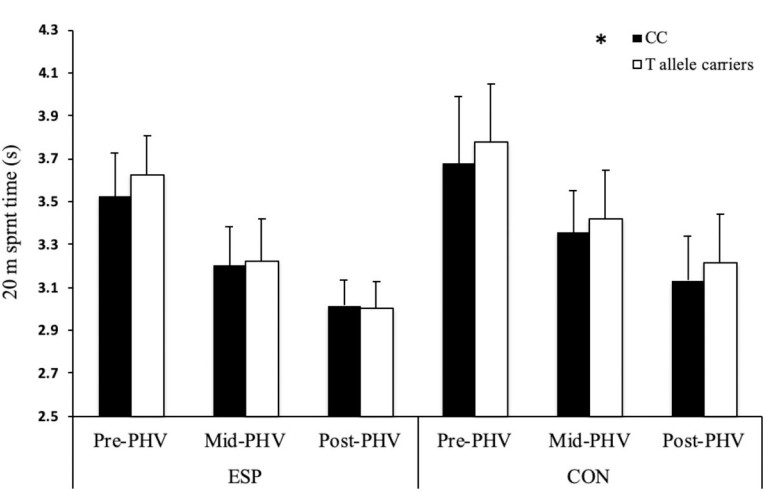

**Fig 1. The association between *BDNF* (rs6265) and horizontal power, acceleration and sprint capabilities.** The association between *BDNF* (rs6265) C>T genotype and: (a) bilateral horizontal forward countermovement jump (BH CMJ), (b) 10 m sprint and (c) 20 m sprint performance in pre-, mid- and post-peak height velocity (PHV) elite youth soccer players (ESP, pre-PHV: n = 121; mid-PHV: n = 37; post-PHV: n = 313) and controls (CON, pre-PHV: n = 44; mid-PHV: n = 30; post-PHV: n = 57). * Difference between *BDNF* CC homozygotes and T-allele carriers ($P < 0.05$).

$P = 0.018$), with pre-PHV ESP GG homozygotes achieving faster 10 m sprint times than pre-PHV ESP A-allele carriers ($P = 0.003$).

 **20 m sprint performance.** There was a main effect of *COL2A1* genotype (F = 6.188, $P = 0.013$; Fig 2B), with CC homozygotes achieving faster 20 m sprint times than T-allele carriers. However, there was also a two-way interaction between *COL2A1* genotype and athlete status (F = 8.735, $P = 0.003$; Fig 2B), with CON CC homozygotes achieving faster 20 m sprint times than T-allele carriers ($P<0.001$). There was an initial two-way interaction between *COL2A1* genotype and PHV status (F = 3.049, $P = 0.048$) but this did not remain significant following FDR correction. Likewise, there was an initial three-way interaction between *PPARA* genotype, ESP status and maturity status (F = 3.034, $P = 0.049$), with pre-PHV CON C-allele carriers being faster than pre-PHV CON GG genotypes (F = 4.511, $P = 0.034$) but this was not significant after FDR correction. There was a main effect of *BDNF* genotype (F = 7.224, $P = 0.007$; Fig 1C), with GG homozygotes achieving faster 20 m sprint times than T-allele carriers. There was a main effect of *COL5A1* genotype (F = 6.188, $P = 0.029$; Fig 3C), with CC homozygotes achieving faster 20 m sprint times than individuals of CT genotype ($P = 0.024$). There was a non-significant tendency for a main effect of *NOS3* genotype (F = 2.584, $P = 0.076$), with post hoc analyses revealing a tendency for TT homozygotes to achieve faster 20 m sprint times compared to CC genotypes ($P = 0.079$). There was a three-way interaction between *AMPD1* genotype, athlete status and maturity status (F = 4.292; $P = 0.014$), with pre-PHV ESP GG homozygotes achieving faster 20 m sprint times than pre-PHV ESP A-allele carriers ($P = 0.019$).

 **Agility (change of direction) performance.** After FDR correction, there were no significant associations between any SNP and agility 505 performance.

## Correlations between 'performance' total genotype score (TGS) and power/speed performance

 **Determination of 'performance' TGS.** After FDR correction, four SNPs [*AMPD1* G>A (rs17602729), *BDNF* C>T (rs6265), *COL5A1* C>T (rs12722), *COL2A1* C>T (rs2070739)] were associated with various power/sprint performance variables and were, therefore, included in our 'performance' TGS model. The TGS for post-PHV ESP (75.6 ± 13.0) was higher than for CON (73.0 ± 12.2; $P = 0.048$) but did not differ from pre-PHV ESP (73.9 ± 15.1; $P = 0.227$), which in turn did not differ from CON ($P = 0.609$).

## 'Performance' TGS correlations with power/speed performance variables

In pre-PHV ESP, TGS correlated with BV CMJ (r = 0.334, $P<0.001$), BH CMJ (r = 0.272, $P = 0.004$), 10 m acceleration (r = -0.238, $P = 0.012$), 20 m sprint (r = -0.286, $P = 0.002$) and 505 agility score (r = -0.258, $P = 0.017$). In post-PHV ESP, TGS correlated solely with BH CMJ (r = 0.140, $P = 0.014$). In post-PHV CON, TGS correlated with 10 m sprint (r = -0.465, $P<0.001$), 20 m sprint (r = -0.298, $P = 0.029$) and 505 agility (r = -0.418, $P = 0.047$). When ESP and CON were analysed together, pre-PHV TGS correlated with BV CMJ (r = 0.271, $P = 0.001$), BH CMJ (r = 0.195, $P = 0.015$), 10 m sprint (r = -0.210, $P = 0.014$), 20 m sprint (r = -0.242, $P = 0.004$) and 505 agility score (r = -0.258, $P = 0.017$). In all post-PHV participants,

**A**

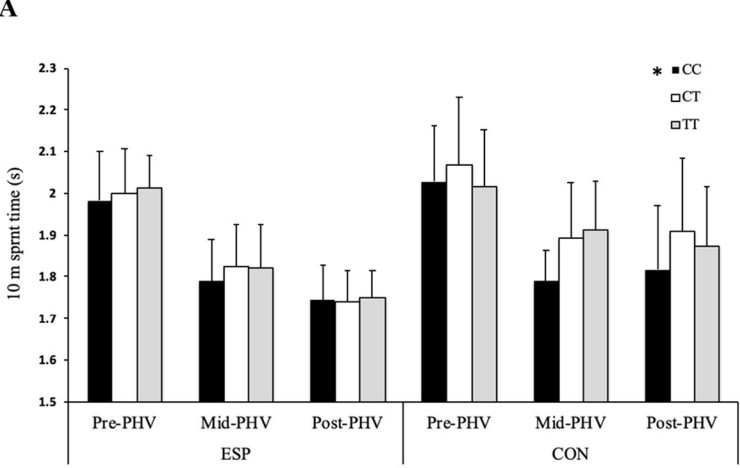

**B**

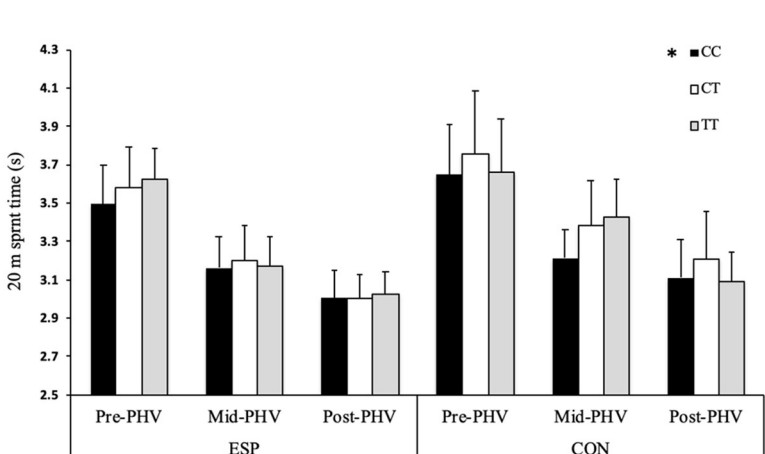

**C**

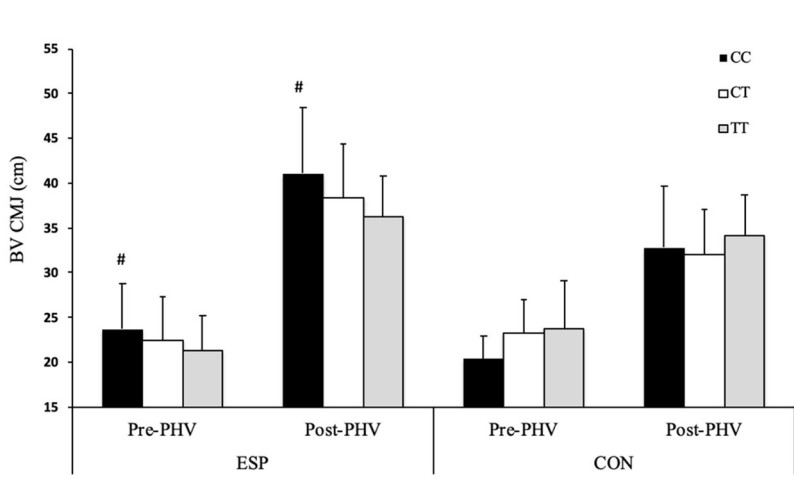

**Fig 3. The association between *COL5A1* (rs12722) and acceleration, sprint and vertical power capabilities.** The association between *COL5A1* (rs12722) C>T genotype and: (a) 10 m sprint, (b) 20 m sprint and (c) bilateral vertical countermovement jump (BV CMJ) performance in pre-, mid- and post-peak height velocity (PHV) elite youth soccer players (ESP, pre-PHV: n = 114; mid-PHV: n = 35; post-PHV: n = 309) and controls (CON, pre-PHV: n = 44; mid-PHV: n = 30; post-PHV: n = 56). * Main effect of *COL5A1* ($P < 0.05$), with a difference between CC and CT genotypes ($P < 0.05$); # difference between ESP CC and TT homozygotes ($P < 0.05$).

**A**

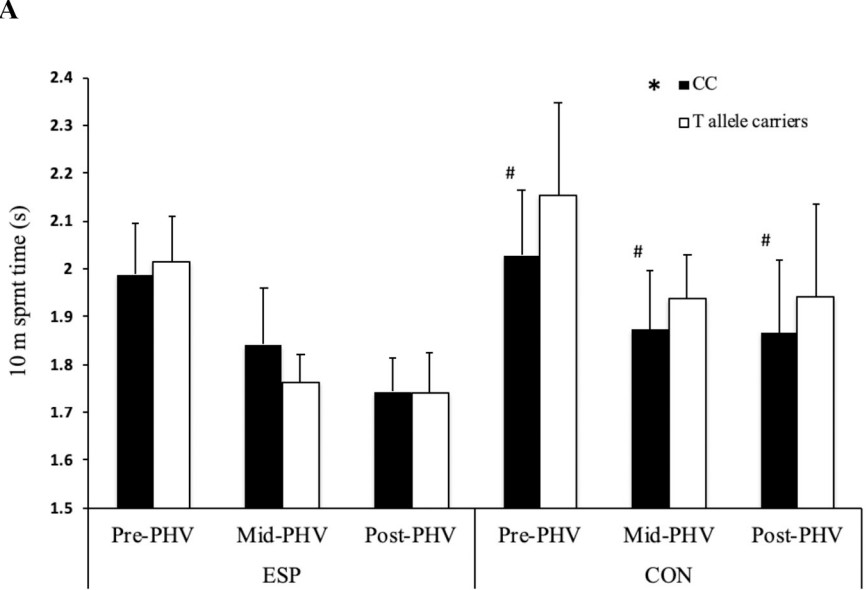

**B**

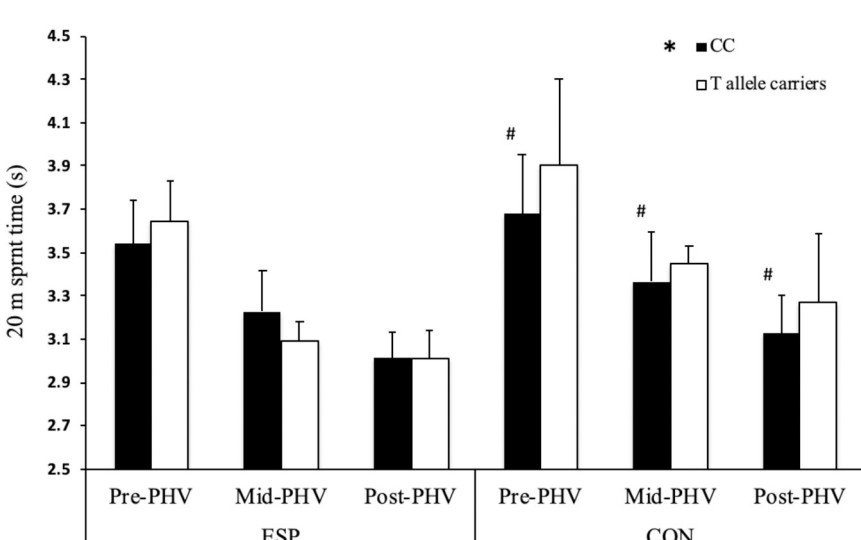

**Fig 2. The association between *COL2A1* (rs2070739) and acceleration and sprint capabilities.** The association between *COL2A1* (rs2070739) C>T genotype and: (a) 10 m sprint and (b) 20 m sprint performance in pre-, mid- and post-peak height velocity (PHV) elite youth soccer players (ESP, pre-PHV: n = 121; mid-PHV: n = 37; post-PHV: n = 314) and controls (CON, pre-PHV: n = 44; mid-PHV: n = 29; post-PHV: n = 57). * Difference between *COL2A1* CC homozygotes and T-allele carriers (*P* < 0.05); # difference between CON CC homozygotes and T-allele carriers (*P* < 0.05).

TGS correlated with BH CMJ (r = 0.198, *P* = 0.018) and BV CMJ (r = 0.134, *P* = 0.010). Regarding muscle physiological variables in post-PHV ESP and CON (*n* = 42), the 'performance' TGS correlated with quadriceps femoris (QF) muscle volume (*r* = 0.422; *P* = 0.006), QF PCSA (*r* = 0.425; *P* = 0.006) and knee extension MVC (*r* = 0.388; *P* = 0.012). In post-PHV

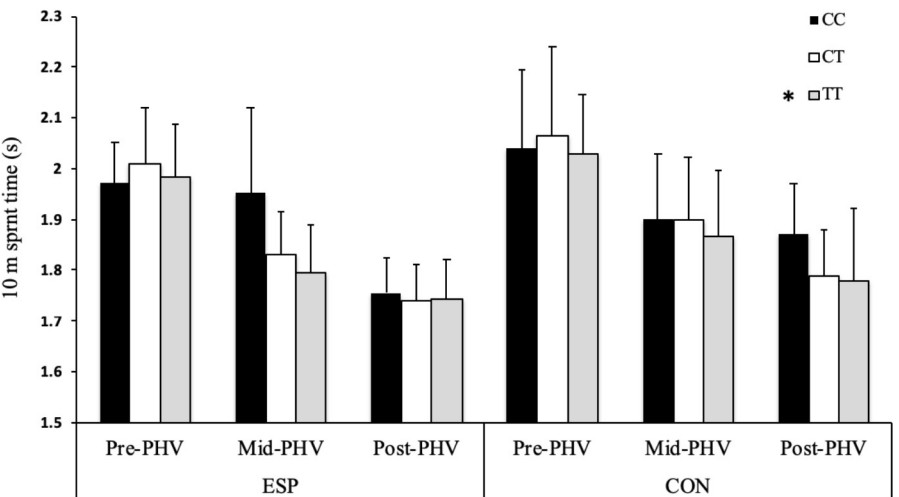

**Fig 4. The association between *NOS3* (rs2070744) and acceleration capability.** The association between *NOS3* (rs2070744) C>T genotype and 10 m acceleration in pre-, mid- and post-peak height velocity (PHV) elite youth soccer players (ESP, pre-PHV: n = 121; mid-PHV: n = 36; post-PHV: n = 313) and controls (CON, pre-PHV: n = 44; mid-PHV: n = 30; post-PHV: n = 57). * Main effect of *NOS3* genotype (*P* < 0.05), with a difference between TT and CC homozygotes (*P* < 0.05).

ESP only (*n* = 22), the 'performance' TGS correlated with QF muscle volume (*r* = 0.573; *P* = 0.005) and QF PCSA (*r* = 0.566; *P* = 0.006).

## SNPs associated with skeletal muscle physiological variables

In a sub-sample of 42 post-PHV participants (ESP, *n* = 22; and CON, *n* = 20), we found associations between three SNPs [*AGT* A>G (rs699), *COL5A1* C>T (rs12722) and *PPARA* G>C

**Table 4. Quadriceps femoris muscle strength, size and architecture in 42 post-PHV soccer players (ESP, *n* = 22; and CON, *n* = 20).** Mean ± SD data are presented according to genotype for three different SNPs.

| SNP | Quadriceps femoris muscle physiological characteristic | | | |
|---|---|---|---|---|
| | **KE MVC (N·m)** | **QF $V_m$ (L)** | **QF PCSA (cm$^2$)** | **VL $\theta_p$ (°)** |
| *AGT* **A>G (rs699)** | | | | |
| AA (*n* = 6) | 252 ± 40 | 2.5 ± 0.2 | 203 ± 15 | 12.8 ± 1.3 |
| AG (*n* = 23) | 306 ± 61 | 2.5 ± 0.3 | 197 ± 31 | 14.9 ± 1.8 |
| GG (*n* = 13) | 349 ± 71** | 3.0 ± 0.6*** | 235 ± 48** | 15.3 ± 2.5* |
| *COL5A1* **C>T (rs12722)** | | | | |
| CC (*n* = 17) | 345 ± 70** | 2.8 ± 0.6* | 225 ± 43* | 14.8 ± 2.3 |
| CT (*n* = 18) | 297 ± 61 | 2.6 ± 0.3 | 207 ± 32 | 14.7 ± 2.1 |
| TT (*n* = 7) | 268 ± 46 | 2.4 ± 0.2 | 180 ± 26 | 14.6 ± 2.0 |
| *PPARA* **G>C (rs4253778)** | | | | |
| GG (*n* = 25) | 293 ± 61 | 2.6 ± 0.4 | 208 ± 34 | 14.4 ± 1.9 |
| GC (*n* = 13) | 343 ± 76 | 2.8 ± 0.5 | 204 ± 43 | 14.3 ± 1.9 |
| CC (*n* = 4) | 329 ± 59 | 2.9 ± 0.7 | 241 ± 47 | 17.9 ± 1.8*** |

*KE MVC*, knee extension maximum voluntary contraction; *QF $V_m$*, quadriceps femoris muscle volume; *PCSA*, physiological cross-sectional area; *VL $\theta_p$*, vastus lateralis muscle fascicle pennation angle

*significantly greater than other genotypes for that respective SNP (*P* < 0.05)

**significantly greater than other genotypes for that respective SNP (*P* < 0.02)

***significantly greater than other genotypes for that respective SNP (*P* < 0.01).

(rs4253778)] and quadriceps femoris muscle volume, PCSA, architecture and maximum strength (Table 4).

## Discussion

The aims of this study were to investigate the association of 10 candidate SNPs with elite *youth* soccer player (ESP) status at different stages of maturity, and to investigate the individual and combined associations of those 10 SNPs with power/speed performance in ESP and recreationally active control (CON) participants, according to maturity status. In a cohort of 686 healthy, young males, including 535 ESP and 151 CON, we have shown for the first time that pre- and post-PHV ESP have distinct polygenic profiles. Post-PHV ESP had a genetic profile that favours superior power/speed characteristics, while pre-PHV ESP had a more endurance-associated polygenic profile. Furthermore, we found that five SNPs were associated with power/speed in ESP at different stages of maturity. The combination of these five SNPs explained a proportion of the variability in all performance tasks in *pre*-PHV ESP, but only the variability in bilateral horizontal-forward countermovement jump (BH CMJ) performance in *post*-PHV ESP. Together, these genetic differences suggest the soccer talent identification procedures (i.e. the process of recognising and recruiting individuals with the perceived potential to become elite players [49,50]) employed by the multiple academies in this study recruit pre- and post-PHV ESP based on different physiological criteria, i.e. endurance (pre-PHV) *vs.* power/speed (post-PHV) capability. Given the importance of power/speed in determining *post*-PHV ESP status [2,9,13,51,52], these findings have significant implications for *future* youth soccer talent identification and development programmes (i.e. intervention programmes prescribed to provide the most optimal environment to allow the player to realise their full potential [49,50]).

We have previously shown that horizontal and vertical power are indicators of ESP playing status in post-PHV ESP but not in pre-PHV ESP [2] but it was not known whether genetic variation could explain these disparities between maturity groups. In the present study, post-PHV ESP demonstrated a greater frequency distribution of the *PPARA* (rs4253778) C-allele and the *AGT* (rs699) G-allele compared to pre-PHV ESP and CON. When maturity groups were combined, we found that ESP had a greater frequency distribution of *PPARA* (rs4253778) C-allele carriers, the *AGT* (rs699) G-allele and the *NOS3* (rs2070744) T-allele compared to CON. Crucially, all three of these alleles have previously been associated with elite power athlete status [15,16,53] and superior power performance [15,16,53,54]. Thus, the higher frequency of the *PPARA* (rs4253778) C-allele and the *AGT* (rs699) G-allele in post-PHV ESP compared to pre-PHV ESP and CON could help explain these previously reported maturation-dependent differences in power/speed between ESP and CON.

The *NOS3* T-allele may exert its influence on ESP and power athlete status via increased gene promoter activity and higher endothelial nitric oxide synthesis [14]. This would stimulate muscle hypertrophy through nitric oxide-mediated vasodilatation [14], thus increasing strength and power. The *AGT* (rs699) G-allele, on the other hand, is thought to be associated with elite power athlete status [15,16,53] via its ability to increase serum angiotensinogen (AGT) concentration [16]. Increased AGT upregulates angiotensin II, which acts as a growth factor in skeletal muscle, and can modulate skeletal muscle hypertrophy in response to mechanical loading, thus enhancing power/speed [14,16,53]. The *PPARA* (rs4253778) C-allele has previously been associated with ESP status [3,4] and a greater composition of type II skeletal muscle fibres [18], which are larger (therefore producing more force) and able to shorten quicker (thus able to generate more power) than type I fibres [17]. These findings therefore

support our original hypothesis that post-PHV ESP would demonstrate a polygenic profile more advantageous for power/speed performance.

In line with the above maturity/athlete status-dependent differences in allele/genotype frequency distribution, we found that the frequency distribution of *ACTN3* (rs1815739) TT (XX) genotype was greater in pre-PHV ESP compared to CON (29.0% vs. 15.9%, respectively). *ACTN3* XX homozygotes cannot produce alpha-actinin-3 [24], a structural sarcomeric protein found in type II skeletal muscle fibres that inhibits the transition of larger, stronger, more powerful type II fibres towards a smaller, weaker more oxidative phenotype [23]. This probably explains the greater muscle strength, power and volume observed in R-allele carriers compared to XX homozygotes [55], as well as the higher frequency distribution of the R-allele in elite power athletes compared to control participants and endurance athletes [56]. Thus, the *PPARA* (rs4253778), *AGT* (rs699) and *ACTN3* (rs1815739) allele/genotype frequency distributions within our ESP and CON cohorts suggest that the polygenic profile of pre-PHV ESP favours fatigue resistance, rather than power/speed.

Indeed, when all four SNPs associated with ESP status [*PPARA* (rs4253778), *AGT* (rs699), *ACTN3* (rs1815739) and *NOS3* (rs2070744)] were combined to create a 'ESP' total genotype score (TGS) [46], post-PHV ESP had a higher TGS than both pre-PHV and CON, with no difference between pre-PHV and CON, thus supporting the individual SNP differences in allele/genotype frequency distribution. In English Category 1 soccer academies, players do not compete 11 *vs.* 11 on a full-sized pitch until they reach ~13 years of age (approximately when PHV occurs [40]). While 11 *vs.* 11 match-play requires a greater percentage of time spent at low speeds and more frequent explosive sprint actions, match-play on smaller pitches with fewer players requires a greater distance per minute, more time running at moderate speeds, and a higher work load per minute [57], thus placing a greater physiological demand on aerobic rather than anaerobic capacity. The playing demands of elite soccer therefore differ according to maturity status and, at pre-PHV level, aerobic capacity is more important than power/speed, which would explain the endurance-favourable genetic profile of our pre-PHV ESP. However, at post-PHV level, when powerful actions are performed more frequently [51] and often determine the outcome of games [58], academy recruitment staff may choose to release players with suboptimal power capabilities (and therefore a suboptimal power genetic profile) from their contracts in preference for players with superior power capabilities (and therefore a more favourable power genetic profile). This probably explains the superior power previously observed in ESP *vs.* CON at post-PHV [2,9,13,51,52] but not at pre-PHV [2]. Overall, this novel information suggests that current elite soccer pre-PHV talent identification procedures may benefit from the addition of genetic profiling to increase the chance of developing powerful players from a young age through to first team.

As well as investigating differences in genetic profile between maturity status within ESP, we sought to examine the genetic association with power/speed performance in ESP at different stages of maturity. Four of the five SNPs associated with power/speed in this study [*NOS3* C>T (rs2070744), *COL5A1* C>T (rs12722), *BDNF* C>T (rs6265), *AMPD1* G>A (rs17602729)] have previously been investigated within a sport/exercise context. Interestingly, with the exception of *NOS3* C>T, these were different SNPs to the four that demonstrated different allele/genotype frequency distributions between pre- and post-PHV ESP and CON. Although we found no association between *NOS3* C>T and power *per se*, we are the first to show that the TT genotype was associated with greater acceleration/sprint speed, which is a component of power. The *NOS3* TT genotype may therefore stimulate an increase in the number of sarcomeres arranged in series rather than in parallel, thus increasing muscle fibre length, which is the main determinant of muscle fibre contraction velocity [59].

While muscle fibre length is a determinant of power output, the extensibility of the tendon is related to speed and horizontal-forward power capabilities in sprinters [10] and post-PHV ESP [9], respectively. Our data show that, regardless of maturity and athlete status, individuals of *COL5A1* (rs12722) CC genotype achieved quicker acceleration and sprint times compared to CT heterozygotes. As the CC genotype has previously been associated with more extensible knee extensor tendons [11], it is possible that the tendons of CC homozygotes can store and release more energy, thus amplifying power during sprint and acceleration. Our data also show that ESP CC homozygotes achieved greater BV CMJ height compared to TT homozygotes. Assuming that CC homozygotes have more elastic tendons compared to TT homozygotes [11], our findings are somewhat in agreement with Kubo and colleagues [60], who reported that greater compliance of the VL tendon–aponeurosis complex facilitates improved countermovement jump performance. This relationship may only be significant in ESP, as they could probably co-ordinate the vertical jump actions better, thus gaining greater use of the stretch shortening cycle and elastic tendon properties. Nevertheless, we are the first to show that *COL5A1* (rs12722) CC genotype is associated with superior acceleration/sprint and vertical jump capabilities, presumably due to the regulatory role that procollagen type V has in controlling fibril diameter and patellar tendon elasticity [11]. Furthermore, this is the first time the *COL2A1 C>T* (rs2070739) SNP has been investigated in a sport/exercise context and our data show that CC homozygotes sprinted quicker than their T-allele carriers, possibly due to having a greater capacity to store and release mechanical energy in the tendon, similarly to *COL5A1* CC homozygotes.

In addition to having more compliant patellar tendons [10], sprinters producing more horizontal force during sprinting are able to highly activate their hamstring muscles just before ground contact [19]. Neuromuscular activation is therefore a major factor in determining sprint performance, and neural function is largely determined by neurotrophins, of which BDNF is one of the most active [28]. In this study, we have shown for the first time that *BDNF* (rs6265) CC homozygotes demonstrated greater horizontal power, acceleration and sprint performance. As the C-allele is associated with a greater abundance of exercise-induced serum BDNF concentration [20], our findings suggest that individuals with potentially enhanced neuromuscular characteristics perform better in the performance tasks that require the body to be propelled in the horizontal-forward direction.

We have also shown that *AMPD1* (rs17602729) GG homozygotes achieved quicker acceleration and sprint times in pre-PHV ESP only. As reduced muscle AMPD1 activity can compromise the capacity to deplete ATP pools and accumulate adenosine monophosphate during high intensity exercise [61], it is possible that pre-PHV ESP *AMPD1* GG homozygotes can repeat high intensity acceleration and sprint efforts more frequently during soccer match-play. As speed performance prior to the onset of maturation is underpinned by inter- and intra-muscular co-ordination [62], performing a greater volume of such actions may lead to preferential adaptations in acceleration and sprint performance in GG homozygotes compared to their A-allele counterparts. The *AMPD1* G>A SNP is the only one of our panel of 10 SNPs that appears to influence acceleration and speed in pre-PHV ESP on an individual SNP basis.

Individual SNP associations with power/speed in ESP can potentially elucidate novel mechanisms linking genotype with tissue phenotype with physical performance. However, these complex power/speed tasks are underpinned by a combination of different physiological factors [9,13], which in turn are thought to be influenced by a combination of several genetic variants [53]. Therefore, we developed a novel 'performance' polygenic profile based on only those five key SNPs shown to be associated with acceleration, speed, vertical and/or horizontal power capabilities on an individual SNP basis. In pre-PHV ESP, there were five significant but weak correlations between the 'performance' TGS and acceleration, sprint, vertical power,

horizontal power and agility performance, while in post-PHV ESP, there was only one positive, but very weak, correlation between the 'performance' TGS and horizontal power performance. The fact that these correlations were weak was not unexpected, as the contribution of single gene variants to relatively complex powerful tasks was always likely to be small. The higher number of correlations in pre- *vs*. post-PHV ESP could be due to the ESP environment increasingly impacting on their physical performance as they mature (e.g. training exposure in ESP increases and includes more athletic development and resistance training as well as soccer activities [63]). Furthermore, practitioners may attempt to improve the power/speed capabilities of less powerful post-PHV ESP through personalised training, which may help compensate to some degree for an unfavourable power/speed genetic profile, thus reducing the likelihood of finding relationships between this particular TGS and power/speed performance in post-PHV ESP.

We have also shown that, while not one single SNP was associated with agility performance, a combination of favourable power/speed genotypes may predispose pre-PHV ESP to a greater ability to change direction quickly. As agility performance has been associated with co-ordination and unilateral reactive strength capabilities [64], the main physiological factors underpinning agility capacity appear to be a combination of neural (inter-muscular co-ordination, neural firing frequency and motor unit synchronization) and muscle-tendon unit mechanical properties. It therefore seems logical that agility is determined by a combination of SNPs, rather than one individual genetic variant. Overall, our novel polygenic analysis shows that, while one SNP may not have a significant impact on explosive performance, a combination of SNPs could have a favourable effect, especially in pre-PHV individuals.

We acknowledge some limitations with our study. For example, we did not measure the physiological determinants of power/speed (e.g. lower limb muscle-tendon properties) in all of our participants. However, these SNPs were investigated on the basis of our previous findings that power [52], muscle morphology [13] and tendon properties [9] differed between post-PHV ESP and CON, and that muscle [13] and tendon [9] properties are related to power generated during CMJs in different directions. Measuring muscle-tendon properties at the cellular and/or molecular level would have required harvesting muscle/tendon biopsies, which would have been highly invasive and impractical in such a large cohort. It is possible, however, to measure muscle-tendon properties non-invasively *in vivo* using ultrasonography and isokinetic dynamometry [9,13]. Although it was not feasible to perform these additional assessments in all 686 participants, we completed them on a sub-sample of *n* = 42 (22 post-PHV ESP and 20 post-PHV CON; Table 4). We found significant main effects of *COL5A1*, *AGT*, and *PPARA* genotypes, i.e. those genotypes associated with superior power/speed phenotypes from the main study data were also associated with greater quadriceps muscle volume and physiological cross-sectional area (PCSA), muscle fascicle pennation angle (an indirect indicator of muscle fibre CSA) and/or greater maximum knee extensor strength, all physiological variables that are related to power [9,13]. Furthermore, both our 'ESP' and 'performance' TGSs correlated with quadriceps volume, which is the product of muscle PCSA (the main determinant of force) multiplied by muscle fascicle length (the main determinant of muscle contraction velocity). As power is the product of force x velocity, these correlations suggest that the combination of these nine SNPs influence power/speed, at least in part, via their combined influence on muscle volume and/or PCSA. Thus, although the muscle physiological phenotype-genotype associations were completed on a very limited sample size (*n* = 42), and should therefore be treated with caution, they do provide physiological evidence for the genotype-power/speed performance associations observed in our much larger cohort of ESP and CON (*n* = 686). Secondly, our estimation of PHV was not a direct measure of pubertal status, so we cannot make direct assertions regarding genetic associations with pubescent phase. However, previous

studies have shown that PHV estimated via the methods employed in this study does compare well with direct measures of pubertal growth [40]. Moreover, measuring pubertal status directly involves invasive procedures that would not be practically feasible to undertake in 686 pre-, mid- and post-PHV males. Finally, both ESP and CON comprised participants of different ethnic origin, which may be considered a limitation due to evidence that genetic variation can differ between ethnic groups [65]. However, we sought to include as many ESP as possible, regardless of ethnicity, so that our results would more accurately reflect all ESP in England and Uruguay, rather than only those of a particular ethnic group. Nevertheless, despite ethnicity information only being available in 38% of CON, sub-sample analysis in participants of known Caucasian origin (ESP, $n = 422$; CON, $n = 54$) revealed that *ACTN3* XX genotype frequency distribution was higher ($P = 0.004$) in pre-PHV ESP (32.0%) compared to both CON (13.0%) and post-PHV ESP (20.4%). This therefore supports our total cohort data.

To conclude, in a cohort of 535 ESP (plus 151 CON) from multiple academies across two continents (thus raising the external validity of our study), we have shown for the first time that pre- and post-PHV ESP have distinct genetic profiles thought to favour endurance and power/speed capabilities, respectively. We have also demonstrated that power, acceleration and sprint performance were associated with five SNPs, both individually and in combination, possibly by influencing muscle size and neuromuscular activation. Therefore, our results not only identify potentially novel physiological mechanisms underpinning the influence of muscle-tendon properties on power/speed performance, but they suggest that pre-PHV ESP are currently recruited with a genetic profile more suited for endurance rather than power and speed, with the latter phenotypes being crucial attributes for post-PHV ESP. Elite soccer academies may wish to consider this information when recruiting talented pre-PHV players, with a view of developing those players for the different demands of the post-PHV game, rather than relying on the recruitment of new players at that stage of maturity. Furthermore, before genetic testing can be considered a potentially useful tool to assist in talent identification and development, replication of our results by independent groups and the identification of more genetic markers of performance in youth soccer is necessary.

## Supporting information

**S1 File.**
(XLSX)

## Acknowledgments

We are extremely grateful to the following individuals for their assistance with data collection: Ignacio Turrens, Ignacio Berriel, Maximiliano Bizzio, Gabriel González, Guillermo Gil, Santiago Ferro, Nicolas Silva, Cesar Santos, Fernando Fadeuille, Lucia Alza, Barbara Buriani, Florencia Fraque, Luciana Tripodi, Melissa Nuñez, Valeria Acosta, Federico Vanni, Cecilia Mariño, Carolina Mutilva, Carolina Pintos, Robert Naughton, Matthew Girven, Stephen McQuilliam, and Raffaella Rodighiero.

## Author Contributions

**Conceptualization:** Conall F. Murtagh, Robert M. Erskine.

**Data curation:** Conall F. Murtagh, Robert M. Erskine.

**Formal analysis:** Conall F. Murtagh, Robert M. Erskine.

**Funding acquisition:** Barry Drust, Robert M. Erskine.

**Investigation:** Conall F. Murtagh, Thomas E. Brownlee, Sebastian Roquero, Sacha Moreno, Giovani Lugioratto, Philipp Baumert, Daniel C. Turner, Dongsun Lee, Peter Dickinson, K. Amber Lyon, Bahare Sheikhsaraf, Betül Biyik, Robert M. Erskine.

**Methodology:** Conall F. Murtagh, Robert M. Erskine.

**Project administration:** Conall F. Murtagh, Thomas E. Brownlee, Edgardo Rienzi, Sebastian Roquero, Sacha Moreno, Robert M. Erskine.

**Resources:** Conall F. Murtagh, Edgardo Rienzi, Sebastian Roquero, Sacha Moreno, Gustavo Huertas, Giovani Lugioratto, Andrew O'Boyle, Ryland Morgans, Andrew Massey, Barry Drust, Robert M. Erskine.

**Software:** Conall F. Murtagh, Sebastian Roquero, Gustavo Huertas, Philipp Baumert, Daniel C. Turner, Dongsun Lee, Peter Dickinson, K. Amber Lyon, Bahare Sheikhsaraf, Betül Biyik, Andrew O'Boyle, Ryland Morgans, Andrew Massey, Robert M. Erskine.

**Supervision:** Barry Drust, Robert M. Erskine.

**Validation:** Conall F. Murtagh, Robert M. Erskine.

**Visualization:** Conall F. Murtagh, Robert M. Erskine.

**Writing – original draft:** Conall F. Murtagh, Robert M. Erskine.

**Writing – review & editing:** Conall F. Murtagh, Thomas E. Brownlee, Edgardo Rienzi, Sebastian Roquero, Sacha Moreno, Gustavo Huertas, Giovani Lugioratto, Philipp Baumert, Daniel C. Turner, Dongsun Lee, Peter Dickinson, K. Amber Lyon, Bahare Sheikhsaraf, Betül Biyik, Andrew O'Boyle, Ryland Morgans, Andrew Massey, Barry Drust, Robert M. Erskine.

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
