## [Decision Letter · Decision Letter 0]

19 Nov 2019

PONE-D-19-26674

THE GENETIC PROFILE OF ELITE YOUTH SOCCER PLAYERS AND ITS ASSOCIATION WITH POWER AND SPEED DEPENDS ON MATURATION STATUS

PLOS ONE

Dear Dr Erskine,

Thank you for submitting your manuscript to PLOS ONE. After careful consideration, we feel that it has merit but does not fully meet PLOS ONE’s publication criteria as it currently stands. Therefore, we invite you to submit a revised version of the manuscript that addresses the points raised during the review process.

Please, pay particular (but not limitedly to) attention to Reviewer 1's issues.

We would appreciate receiving your revised manuscript by Jan 03 2020 11:59PM. To enhance the reproducibility of your results, we recommend that if applicable you deposit your laboratory protocols in protocols.io, where a protocol can be assigned its own identifier (DOI) such that it can be cited independently in the future. For instructions see: http://journals.plos.org/plosone/s/submission-guidelines#loc-laboratory-protocols

We look forward to receiving your revised manuscript.

Kind regards,

Luca Paolo Ardigò, Ph.D.

Academic Editor

PLOS ONE

Journal Requirements:

1. We note that you have included the phrase “data not shown” in your manuscript. Unfortunately, this does not meet our data sharing requirements. PLOS does not permit references to inaccessible data. We require that authors provide all relevant data within the paper, Supporting Information files, or in an acceptable, public repository. Please add a citation to support this phrase or upload the data that corresponds with these findings to a stable repository (such as Figshare or Dryad) and provide and URLs, DOIs, or accession numbers that may be used to access these data. Or, if the data are not a core part of the research being presented in your study, we ask that you remove the phrase that refers to these data.

Additional Editor Comments (if provided):

Please, pay particular (but not limitedly to) attention to Reviewer 1's issues.

Reviewers' comments:

Reviewer's Responses to Questions

**Comments to the Author**

1. Is the manuscript technically sound, and do the data support the conclusions?

Reviewer #1: Yes

Reviewer #2: Yes

Reviewer #3: Yes

2. Has the statistical analysis been performed appropriately and rigorously? 

Reviewer #1: No

Reviewer #2: Yes

Reviewer #3: Yes

3. Have the authors made all data underlying the findings in their manuscript fully available?

Reviewer #1: Yes

Reviewer #2: Yes

Reviewer #3: Yes

4. Is the manuscript presented in an intelligible fashion and written in standard English?

Reviewer #1: Yes

Reviewer #2: Yes

Reviewer #3: Yes

5. Review Comments to the Author

Reviewer #1: The author studied the association of 10 SNPs and elite athlete status and power/speed performance in 536 youth soccer players and 151 age-matched controls. The main issue of this study is founding by population stratification.

There are more Caucasians in the ESP group vs control group and many of the SNPs being investigated showed very different allele frequencies in Caucasian and African American population (e.g., MAF of rs4253778 in PPARA is 0.19 in Caucasian and 0.95 in African American). This will cause inflated Type I error rate (false positives). The analysis should either adjust for race or stratify on race. Without race information or a genomewide genotype data (to cluster the sample into different ethnic groups) the results are questionable.

Table 2. Please double check all the statistical tests. Some do not seem to be right. For example, rs4253778 in PPARA, neither the comparison of 18.8% vs 25.5% nor 25.5% vs 18.9% had p-value <0.05. Some SNPs a dominant model was assumed but some were not. Is that purely based on an allele frequency cutoff?

Table 2. When there are only two categories of a variable only one category needs to be shown in the table since the second categories carries identical information. For example, for C vs G alleles or CC/GG vs GG genotypes of PPARA rs4253778, only one allele and one genotype group need to be shown.

Table 2. Please show actual p-values. Please incorporate rs number in the table

The authors mentioned FDR to correct for multiple testing but I don’t find FDR results in the paper. It is unclear how FDR were used since there are many statistical tests in this paper, over all p-values?

It is mentioned that “three-way between-subjects ANOVA was used to investigate the association between genotype…”. This is will unbalance design. It is not clear what terms were included in the anova, two-way interaction? three-way interaction?

Page 17, line 295, please double check the p-value.

Reviewer #2: The study is well designed and presented. My only caution are the statements suggesting that the genetic information could be used for talent identification purposes. This is a controversial issue as a result of the claims by may DTC companies. I suggest that the authors focus on the genetic contribution to the studied phenotype.

Reviewer #3: 1. In the abstract, please confirm that the subjects were all male.

2. INTRODUCTION:

• In the context of athlete’s genetic studies more comprehensive review of literature is required.

• There is a lack of information about the function of 10 genes and their polymorphisms. Given that this is the premise of the research, this should be included. For example, a table could be provided describing the functions of genes and SNPs.

3. What do the authors mean with " talent identification and development programmes " ? This point should be clarified.

4. In the Results section, lines 256 and 280, please check the numbering of the tables.

6. PLOS authors have the option to publish the peer review history of their article (what does this mean?). If published, this will include your full peer review and any attached files.

Reviewer #1: No

Reviewer #2: Yes: Malcolm Collins

Reviewer #3: No

---

## [Author Response · Author response to Decision Letter 0]

6 Feb 2020

Dear Editor and Reviewers, 

We are very grateful to you all for you taking the time to review our manuscript and for providing very constructive comments. We believe the changes we have made to our revised manuscript in response to these comments have significantly improved the quality of the manuscript, so thank you. We have considered all comments and responded to each one in turn. 

Best wishes, 

Rob Erskine on behalf of all co-authors

Editor Comments:

1. We note that you have included the phrase “data not shown” in your manuscript. Unfortunately, this does not meet our data sharing requirements. PLOS does not permit references to inaccessible data. We require that authors provide all relevant data within the paper, Supporting Information files, or in an acceptable, public repository. Please add a citation to support this phrase or upload the data that corresponds with these findings to a stable repository (such as Figshare or Dryad) and provide and URLs, DOIs, or accession numbers that may be used to access these data. Or, if the data are not a core part of the research being presented in your study, we ask that you remove the phrase that refers to these data. 

AUTHORS’ RESPONSE: Thank you for raising this point. We feel that these data are important for providing further confidence in our main results. Therefore, we have included descriptions of the relevant physiological measurements in the Materials and Methods section (P 12-13), and the data in the results section (Table 3; P 24), as well as correlation data between the two TGSs and muscle physiological variables on P20 and P24: 

Results section, P24: 

SNPs associated with skeletal muscle physiological variables 

In a sub-sample of 42 post-PHV participants (ESP, n = 22; and CON, n = 20), we found associations between three SNPs [AGT A>G (rs699), COL5A1 C>T (rs12722) and PPARA G>C (rs4253778)] and quadriceps femoris muscle volume, PCSA, architecture and maximum strength (Table 3). 

Table 3. Quadriceps femoris muscle strength, size and architecture in 42 post-PHV soccer players (ESP, n = 22; and CON, n = 20). Mean ± SD data are presented according to genotype for three different SNPs. 

SNP Quadriceps femoris muscle physiological property

 KE MVC (N∙m) QF Vm (L) QF PCSA (cm2) VL θp (�)

AGT A>G (rs699)

TT (n = 6) 252 ± 40 2.5 ± 0.2 203 ± 15 12.8 ± 1.3 

TC (n = 23) 306 ± 61 2.5 ± 0.3 197 ± 31 14.9 ± 1.8

CC (n = 13) 349 ± 71** 3.0 ± 0.6*** 235 ± 48** 15.3 ± 2.5*

COL5A1 C>T (rs12722)

CC (n = 17) 345 ± 70** 2.8 ± 0.6* 225 ± 43* 14.8 ± 2.3

CT (n = 18) 297 ± 61 2.6 ± 0.3 207 ± 32 14.7 ± 2.1

TT (n = 7) 268 ± 46 2.4 ± 0.2 180 ± 26 14.6 ± 2.0

PPARA G>C (rs4253778)

GG (n = 25) 293 ± 61 2.6 ± 0.4 208 ± 34 14.4 ± 1.9

GC (n = 13) 343 ± 76 2.8 ± 0.5 204 ± 43 14.3 ± 1.9

CC (n = 4) 329 ± 59 2.9 ± 0.7 241 ± 47 17.9 ± 1.8***

KE MVC, knee extension maximum voluntary contraction; QF Vm, quadriceps femoris muscle volume; PCSA, physiological cross-sectional area; VL θp, vastus lateralis muscle fascicle pennation angle; *significantly greater than other genotypes for that respective SNP (P < 0.05); **significantly greater than other genotypes for that respective SNP (P < 0.02); ***significantly greater than other genotypes for that respective SNP (P < 0.01). 

The ‘ESP’ TGS correlation data on P20: 

… “Regarding muscle physiological variables in post-PHV ESP and CON (n = 42), the ‘ESP’ TGS correlated with quadriceps femoris (QF) muscle volume (r = 0.400; P = 0.010), QF PCSA (r = 0.334; P = 0.033), VL θp (r = 0.416; P = 0.007) and knee extension MVC (r = 0.408; P = 0.008). In post-PHV ESP only (n = 22), the ‘ESP’ TGS correlated with QF muscle volume (r = 0.430; P = 0.046) and QF PCSA (r = 0.438; P = 0.041), VL θp (r = 0.441; P = 0.040) and knee extension MVC (r = 0.474; P = 0.026).”

The ‘performance’ TGS correlation data on P24: 

Regarding muscle physiological variables in post-PHV ESP and CON (n = 42), the ‘performance’ TGS correlated with quadriceps femoris (QF) muscle volume (r = 0.422; P = 0.006), QF PCSA (r = 0.425; P = 0.006) and knee extension MVC (r = 0.388; P = 0.012). In post-PHV ESP only (n = 22), TGS correlated with QF muscle volume (r = 0.573; P = 0.005) and QF PCSA (r = 0.566; P = 0.006).

2. Please, pay particular (but not limitedly to) attention to Reviewer 1's issues.

AUTHORS’ RESPONSE: Thank you. Once again, we are very grateful to have received such constructive comments from all three reviewers, and we have responded to all comments raised.

Reviewers' comments:

Reviewer's Responses to Questions

Comments to the Author

1. Is the manuscript technically sound, and do the data support the conclusions?

Reviewer #1: Yes

Reviewer #2: Yes

Reviewer #3: Yes

2. Has the statistical analysis been performed appropriately and rigorously?

Reviewer #1: No

AUTHORS’ RESPONSE: We have responded more comprehensively to Reviewer #1’s comments regarding our statistical approach in section 5, below. We believe our statistical analyses have been performed appropriately and extremely rigorously. This may not have been apparent in our original submission due to our desire to be succinct but, in response to Reviewer #1’s requests, we have included comprehensive information surrounding our statistical approaches in our revised manuscript, which should demonstrate the appropriateness and rigour of our statistical analyses. 

Reviewer #2: Yes

Reviewer #3: Yes

3. Have the authors made all data underlying the findings in their manuscript fully available?

Reviewer #1: Yes

Reviewer #2: Yes

Reviewer #3: Yes

4. Is the manuscript presented in an intelligible fashion and written in standard English?

Reviewer #1: Yes

Reviewer #2: Yes

Reviewer #3: Yes

5. Review Comments to the Author

Reviewer #1: The author studied the association of 10 SNPs and elite athlete status and power/speed performance in 536 youth soccer players and 151 age-matched controls. The main issue of this study is founding by population stratification. There are more Caucasians in the ESP group vs control group and many of the SNPs being investigated showed very different allele frequencies in Caucasian and African American population (e.g., MAF of rs4253778 in PPARA is 0.19 in Caucasian and 0.95 in African American). This will cause inflated Type I error rate (false positives). The analysis should either adjust for race or stratify on race. Without race information or a genomewide genotype data (to cluster the sample into different ethnic groups) the results are questionable.

AUTHORS’ RESPONSE: Thank you for raising this point. In the discussion of our originally submitted manuscript, we acknowledged that the inclusion of different ethnic groups may be regarded as a limitation of our study due to these points mentioned by Reviewer #1 (although the data we have seen from the 1000 genome project demonstrates a slightly smaller difference in rs4253778 MAF, e.g. 0.192 vs. 0.745 for EUR vs. AFR). 

Although the absolute number of Caucasians was higher in ESP compared to CON (the overall group size of ESP was larger than CON, regardless of ethnic group), the relative proportions of Caucasians (and other ethnic groups) within ESP and the sub-sample of CON were similar. Thus, a comparison between two groups (ESP and CON) with similar relative ethnic group composition should enable a fair comparison. As the ANOVA model compares means between groups, it is unclear to us how our current analysis would have led to type I errors. 

We accounted for the imbalance in the cohort sizes between ESP and CON by ensuring that all data were checked for violations of specific three-way ANOVA assumptions that may invalidate the interpretation of results. More specifically, Mauchly's test of sphericity, which assesses whether the variance of the differences between levels of the within-subject factors are equal, was applied to establish if the homogeneity of variance principle had been violated. If Mauchly's test of sphericity was violated and the variance of the differences between levels of the within-subject factors were not equal, the Greenhouse-Geisser correction method was applied [1]. This correction method therefore accounted for the imbalance in cohort sizes when analysing three-way ANOVAs. In addition, box plots were used to check if there were any outliers (none were found); and all data sets were tested for normality using the Kolmogorov-Smirnov test. Testing for such assumptions allowed us to be confident that the interpretation of the three-way ANOVAs was robust despite different sample sizes in CON and ESP.

We have added the following to the statistical analysis section of the manuscript on P16:

We accounted for the imbalance in the cohort sizes between ESP and CON by ensuring that all data was checked for violations of specific three-way ANOVA assumptions that may invalidate the interpretation of results. More specifically, Mauchly's test of sphericity, which assesses whether the variance of the differences between levels of the within-subject factors are equal, was applied to establish if the homogeneity of variance principle had been violated. If Mauchly's test of sphericity was violated, the Greenhouse-Geisser correction method was applied [1]. To test for other assumptions, box plots were used to check if there were any outliers (none were found); and all data sets were tested for normality using the Kolmogorov-Smirnov test.

Finally, as discussed in our originally submitted manuscript, elite soccer academies in the countries included in our study do not comprise athletes from solely one ethnic group (as demonstrated from our ethnicity data), and we felt it was important that our results were representative of actual soccer academies rather than one ethnic group. Thus, we aimed to recruit our CON group with a similar composition of ethnic groups to our ESP cohort. Further, if we were able to stratify according to ethnic group, our sample sizes (which are further broken down into different maturation and genotype groups), particularly for non-Caucasian participants, would become significantly smaller, thus reducing the statistical power of our study and increasing the risk of making type II errors, i.e. finding false negatives. This would also make the study design extremely complicated. Critically, however, we do not have ethnic information for all of our control group (as originally stated in Materials and Methods), so stratifying according to ethnic group with a CON group sample size of n=59 (which would be further broken down into different maturation and genotype groups) would reduce statistical power even further. Therefore, although the reviewer raises a valid point (which we have acknowledged in the Discussion), with respect, we maintain that including ethnic group as a co-variate in a three-way between groups ANCOVA model, or analysing our data separately, according to ethnic group, would undoubtedly lead to type II errors. 

Table 2. Please double check all the statistical tests. Some do not seem to be right. For example, rs4253778 in PPARA, neither the comparison of 18.8% vs 25.5% nor 25.5% vs 18.9% had p-value <0.05. Some SNPs a dominant model was assumed but some were not. Is that purely based on an allele frequency cutoff?

AUTHORS’ RESPONSE: Thank you for raising this point. We had actually reported significant differences between the allele frequency distributions you have mentioned in our original submission – please see the footnote beneath the original table. In this case, the asterisk (*) represented a significant (P < 0.05) difference between post-PHV ESP and CON, while the ampersand symbol (&) represented a significant (P < 0.05) difference between post-PHV ESP and pre-PHV ESP. However, we apologise that this was not clear. We have replaced the ampersand symbol with a dagger symbol (�) and highlighted the between group differences within each significant �2 model (please see the newly inserted text in the Table 3 header). We hope this now removes any confusion. 

The reason why a dominant model was assumed was indeed based on a low MAF, for which our threshold was 0.25 in Caucasian populations (as this group was the predominant ethnic group in our study). Thus, for any SNP with a MAF <0.25 in a Caucasian population, the relatively low number of participants with the minor allele homozygote genotype was combined with the heterozygotes for that SNP. The following information has now been added to the table header: 

“All SNPs were analysed using a co-dominant model (e.g. AA vs. Aa vs. aa) except when the minor allele frequency (MAF) for a Caucasian population [2] was <0.25, in which case a dominant model (e.g. AA vs Aa+aa) was used instead. P-values are presented for significant �2 tests of homogeneity and between group differences are identified (P < 0.05), i.e. * difference between All ESP and CON; ** difference between post-PHV ESP and CON; *** difference between pre-PHV ESP and CON; � difference between post- and pre-PHV ESP.”

We thank Reviewer #1 for raising these points. The addition of this information certainly helps to clarify between group differences in allele/genotype frequency distribution and why we have pooled some genotypes but not others. 

Table 2. When there are only two categories of a variable only one category needs to be shown in the table since the second categories carries identical information. For example, for C vs G alleles or CC/GG vs GG genotypes of PPARA rs4253778, only one allele and one genotype group need to be shown.

AUTHORS’ RESPONSE: We are unsure of what Reviewer #1 is referring to with this point, as C vs G is not the same as CC/GG vs GG (or even CC/CG vs GG). This can be demonstrated by visually comparing allele frequency with genotype frequency distribution within the same group for any given SNP. Allele frequency distribution was analysed separately to genotype frequency distribution for each SNP, hence why we reported both the allele and genotype frequency distributions, i.e. we did not compare allele frequency with genotype frequency, so we are unsure why we would present only one allele and one genotype group. The way we have chosen to present these data is a common method of reporting allele and genotype frequency distributions for multiple SNPs. 

Table 2. Please show actual p-values. Please incorporate rs number in the table

The authors mentioned FDR to correct for multiple testing but I don’t find FDR results in the paper. It is unclear how FDR were used since there are many statistical tests in this paper, over all p-values?

It is mentioned that “three-way between-subjects ANOVA was used to investigate the association between genotype…”. This is will unbalance design. It is not clear what terms were included in the anova, two-way interaction? three-way interaction?

Page 17, line 295, please double check the p-value.

AUTHORS’ RESPONSE: Thank you for raising these points. The reason we did not originally include P values in the table was to avoid including too much information, which may confuse the reader. However, we have now reformatted the table to accommodate the reviewer’s request and we agree that the addition of P values to the table is more informative. 

The rs numbers were originally reported in the table heading but we have moved these to the appropriate places within the actual table, as the reviewer has requested. We agree that this avoids duplication of unnecessary information and makes the table clearer. 

An earlier draft of this manuscript included more detail surrounding our FDR methods (both for our �2 tests of homogeneity and three-way between subjects ANOVAs) but we decided to remove this in an attempt to make the manuscript more concise. However, we agree that the addition of this information would make the manuscript more informative, so we have now re-instated the relevant information in the Statistics section on P15. This now explains the FDR for the �2 tests of homogeneity, the 3-way ANOVAs (associations between SNP and performance phenotypes) and the one-way ANOVAs for the newly introduced SNP associations with quadriceps femoris muscle physiological characteristics. The FDR thresholds for each SNP P value were calculated according to Benjamini and Hochberg [3].

Regarding the three-way ANOVA interactions and main effects investigated in this study, we focussed on the three-way interaction (genotype x athlete status x maturation), two-way interaction (genotype x athlete status and genotype x maturation status), and main effect of genotype. If a significant three-way interaction occurred, simple simple main effects and simple simple pairwise comparisons with Bonferroni adjustment revealed significant differences. If a significant two-way interaction occurred, simple main effects and pairwise comparisons with Bonferroni adjustment were performed to reveal differences. If a significant genotype main effect existed, Bonferroni post-hoc tests were used to locate the genotype difference in test performance scores. We have included the following in P16: 

Thus, we focussed on the three-way interaction (genotype x athlete status x maturation), two-way interactions (genotype x athlete status and genotype x maturation status), and main effect of genotype. If a significant three-way interaction occurred, simple simple main effects and simple simple pairwise comparisons with Bonferroni adjustment revealed significant differences. If a significant two-way interaction occurred, simple main effects and pairwise comparisons with Bonferroni adjustment were performed to reveal differences. If a significant genotype main effect existed, Bonferroni post-hoc tests were used to locate the genotype difference in test performance scores.

Furthermore regarding the three-way ANOVAs, we accounted for the imbalance in the cohort sizes between ESP and CON by ensuring that all data was checked for violations of specific three-way ANOVA assumptions that may invalidate the interpretation of results. More specifically, Mauchly's test of sphericity, which assesses whether the variance of the differences between levels of the within-subject factors are equal, was applied to establish if the homogeneity of variance principle had been violated. If Mauchly's test of sphericity was violated and the variance of the differences between levels of the within-subject factors were not equal, the Greenhouse-Geisser correction method was applied [1]. This correction method therefore accounted for the imbalance in cohort sizes when analysing three-way ANOVAs. Box plots were used to check for outliers (none were found), and data sets were normally distributed according to the Kolmogorov-Smirnov test. Testing for such assumptions allowed us to be confident that the interpretation of the three-way ANOVA was robust despite different sample sizes in each group.

We have added the following to the statistical analysis section of the manuscript on P16:

We accounted for the larger cohort size in ESP vs. CON by ensuring that all data was checked for violations of specific three-way ANOVA assumptions that may invalidate the interpretation of results. More specifically, Mauchly's test of sphericity, which assesses whether the variance of the differences between levels of the within-subject factors are equal, was applied to establish if the homogeneity of variance principle had been violated. If Mauchly's test of sphericity was violated, the Greenhouse-Geisser correction method was applied [4]. Box plots were used to check for outliers (none were found); and all data sets were normally distributed according to the Kolmogorov-Smirnov test. 

Finally, thank you for pointing out the typo in the P-value concerning lack of difference between pre- and post-PHV ESP in ‘performance’ TGS. This P-value was actually 0.227, not 0.022, which now agrees with the statement that there was no difference between pre-PHV ESP and CON (P = 0.227). We have also highlighted the lack of difference between pre-PHV ESP and CON (P = 0.609). 

Once again, we are grateful to Reviewer #1 for raising all of these important points, and we feel that the changes we have made as a consequence have improved the quality of the manuscript. 

Reviewer #2: The study is well designed and presented. My only caution are the statements suggesting that the genetic information could be used for talent identification purposes. This is a controversial issue as a result of the claims by may DTC companies. I suggest that the authors focus on the genetic contribution to the studied phenotype.

AUTHORS’ RESPONSE: Thank you for raising this important point. As this is the first study of its kind to investigate the genetic association with power and speed in elite youth soccer players (accounting for the potentially confounding factor of maturation), we wanted to highlight that our results provide preliminary evidence for genetic analyses being a potentially useful tool for future talent ID and development (i.e. after our results have been replicated by independent groups), and only in conjunction with traditional methods for talent ID and development. However, we completely agree with the reviewer’s concern about DTCs currently making false claims, e.g. that we are already at the point where genetic testing can be used for talent ID, injury risk, etc. Hence, our use of the words “assist” and “future” in our original concluding sentence to emphasise this point. On reflection and following the reviewer’s comment, we think this could still be misused by DTC companies as ‘current evidence’ that validates their ‘product’. Therefore, to prevent our findings being mis-represented by DTC companies, we have changed our concluding statement to the following: 

“Elite soccer academies may wish to consider this information when recruiting talented pre-PHV players, with a view of developing those players for the different demands of the post-PHV game, rather than relying on the recruitment of new players at that stage of maturation. Furthermore, before genetic testing can be considered a potentially useful tool to assist in talent identification and development, replication of our results by independent groups and the identification of more genetic markers of performance is necessary.” 

We have also revised our concluding statement in the abstract accordingly: 

“To conclude, we have shown for the first time that pre- and post-PHV ESP have distinct genetic profiles, with pre-PHV ESP more suited for endurance, and post-PHV ESP for power and speed (the latter phenotypes being crucial attributes for post-PHV ESP). We have also demonstrated that power, acceleration and sprint performance were associated with five SNPs, both individually and in combination, possibly by influencing muscle size and neuromuscular activation.”

We hope Reviewer #2 agrees that our take home message has been satisfactorily changed to ensure it is unequivocal in terms of our study not supporting the false claims made by DTC companies. 

Reviewer #3: 

1. In the abstract, please confirm that the subjects were all male.

AUTHORS’ RESPONSE: Thank you for highlighting this omission. We have now included the word “male” in the Abstract. 

2. INTRODUCTION:

• In the context of athlete’s genetic studies more comprehensive review of literature is required.

AUTHORS’ RESPONSE: Thank you for raising this point. As the reviewer is aware, the challenge with writing a good introduction is to include a comprehensive review of the literature to provide a solid background to the study, as well as a convincing rationale for the study, all as succinctly as possible. To do this, we decided to focus on the literature pertaining to the genetic association with elite soccer player status/performance. We have also attempted to review the current literature when introducing the SNPs of interest in the Introduction, e.g. regarding the (potential) mechanistic influence of each SNP. However, we agree with Reviewer #3’s comment that more comprehensive information could be provided regarding the SNPs investigated in the current study. Therefore, in response to this and Reviewer #3’s subsequent comment, we have included a NEW Table 2, which provides functional information on each of the 10 SNPs (plus their respective genes and proteins), supported by both classic and recent literature. We have also added the following part of a sentence and reference pertaining to a recent review of the genetic association with power and speed phenotypes [5] on P3: “Over 69 genetic markers have been associated with power athlete status [5]” and the genetic association with power and speed is likely due to specific gene variants influencing protein abundance/expression, which will in turn affect the physiological determinants of speed and power.

• There is a lack of information about the function of 10 genes and their polymorphisms. Given that this is the premise of the research, this should be included. For example, a table could be provided describing the functions of genes and SNPs.

AUTHORS’ RESPONSE: Thank you for raising this point. In our originally submitted manuscript, we introduced each of our investigated SNPs in the Introduction but we agree that more information about the individual SNPs would be beneficial. Consequently, and in response to the reviewer’s excellent suggestion, we have added this information in our NEW Table 2 on P7. In this table, we have included the gene name, the protein product, the alleles and rs number, location, MAF, and SNP type and function (supported by both classic and recent literature). We feel this provides significantly more information about each SNP, so thank you for raising this. 

3. What do the authors mean with " talent identification and development programmes " ? This point should be clarified.

AUTHORS’ RESPONSE: Thank you for raising this point. Talent identification refers to the process of recognising and recruiting individuals with the perceived potential to become elite players. Talent development is a process whereby the player is provided with the optimal learning environment to achieve their full potential. Consequently, a talent development programme is the intervention programme prescribed to provide the most optimal environment to allow the player to realise their full potential [6,7]. We have now clarified “talent identification and development programmes” in the Discussion on P25:

The combination of these five SNPs explained a proportion of the variability in all performance tasks in pre-PHV ESP, but only the variability in bilateral horizontal-forward countermovement jump (BH CMJ) performance in post-PHV ESP. Together, these genetic differences suggest the soccer talent identification procedures (i.e. the process of recognising and recruiting individuals with the perceived potential to become elite players [6,7]) employed by the multiple academies in this study recruit pre- and post-PHV ESP based on different physiological criteria, i.e. endurance (pre-PHV) vs. power/speed (post-PHV) capability. Given the importance of power/speed in determining post-PHV ESP status [8–12], these findings have significant implications for future youth soccer talent identification and development programmes (i.e. intervention programmes prescribed to provide the most optimal environment to allow the player to realise their full potential [6,7]). 

4. In the Results section, lines 256 and 280, please check the numbering of the tables.

AUTHORS’ RESPONSE: Thank you for pointing this out. We noticed that in some instances the COL2A1 and COL5A1 figure references were mixed up and we have now corrected the as COL2A1 is Figure 2 and COL5A1 is figure 3. We have also removed the asterisk from Figure 4 (NOS3 association with 20 m sprint), as this was a non-significant tendency for a main effect.

6. PLOS authors have the option to publish the peer review history of their article (what does this mean?). If published, this will include your full peer review and any attached files.

Do you want your identity to be public for this peer review? For information about this choice, including consent withdrawal, please see our Privacy Policy.

Reviewer #1: No

Reviewer #2: Yes: Malcolm Collins

Reviewer #3: No

References

1. Maxwell SE, Delaney HD, Kelley K. Designing experiments and analyzing data: A model comparison perspective. 3rd ed. New York: Routledge; 2018.

2. Auton A, Abecasis GR, Altshuler DM, Durbin RM, Abecasis GR, Bentley DR, et al. A global reference for human genetic variation. Nature. 2015;526: 68–74. doi:10.1038/nature15393

3. Benjamini Y, Hochberg Y. Controlling the false discovery rate: a practical and powerful approach to multiple testing. J R Stat Soc Ser B. 1995; 289–300. 

4. Maxwell SE, Delaney HD. Designing experiments and analyzing. Lawrence Erlbaum Associates, Hillsdale. Chapters; 2004. 

5. Maciejewska-Skrendo A, Cięszczyk P, Chycki J, Sawczuk M, Smółka W. Genetic Markers Associated with Power Athlete Status. J Hum Kinet. 2019;68: 17–36. doi:10.2478/hukin-2019-0053

6. Williams AM, Reilly T. Talent identification and development in soccer. J Sports Sci. 2000;18: 657–667. 

7. Sarmento H, Anguera MT, Pereira A, Araujo D. Talent identification and development in male football: a systematic review. Sport Med. 2018;48: 907–931. 

8. Murtagh CF, Brownlee TE, O’Boyle A, Morgans R, Drust B, Erskine RM. Importance of Speed and Power in Elite Youth Soccer Depends on Maturation Status. J strength Cond Res. 2018;32: 297–303. doi:10.1519/JSC.0000000000002367

9. Murtagh CF, Stubbs M, Vanrenterghem J, O’Boyle A, Morgans R, Drust B, et al. Patellar tendon properties distinguish elite from non-elite soccer players and are related to peak horizontal but not vertical power. Eur J Appl Physiol. 2018;118: 1737–1749. doi:10.1007/s00421-018-3905-0

10. Murtagh CF, Nulty C, Vanrenterghem J, O’Boyle A, Morgans R, Drust B, et al. The Neuromuscular Determinants of Unilateral Jump Performance in Soccer Players Are Direction-Specific. Int J Sports Physiol Perform. 2018;13: 604–611. doi:10.1123/ijspp.2017-0589

11. Murtagh CF, Naughton RJ, McRobert AP, O’Boyle A, Morgans R, Drust B, et al. A Coding System to Quantify Powerful Actions in Soccer Match Play: A Pilot Study. Res Q Exerc Sport. 2019; 1–10. 

12. Murtagh CF, Vanrenterghem J, O’Boyle A, Morgans R, Drust B, Erskine RM. Unilateral jumps in different directions: a novel assessment of soccer-associated power? J Sci Med Sport. 2017;20: 1018–1023. doi:10.1016/j.jsams.2017.03.016

---

## [Decision Letter · Decision Letter 1]

9 Mar 2020

PONE-D-19-26674R1

THE GENETIC PROFILE OF ELITE YOUTH SOCCER PLAYERS AND ITS ASSOCIATION WITH POWER AND SPEED DEPENDS ON MATURATION STATUS

PLOS ONE

Dear Dr Erskine,

Thank you for submitting your manuscript to PLOS ONE. After careful consideration, we feel that it has merit but does not fully meet PLOS ONE’s publication criteria as it currently stands. Therefore, we invite you to submit a revised version of the manuscript that addresses the points raised during the review process.

Reviewer 1 still raises major issues: please, put all your effort to address them.

We would appreciate receiving your revised manuscript by Apr 23 2020 11:59PM. To enhance the reproducibility of your results, we recommend that if applicable you deposit your laboratory protocols in protocols.io, where a protocol can be assigned its own identifier (DOI) such that it can be cited independently in the future. For instructions see: http://journals.plos.org/plosone/s/submission-guidelines#loc-laboratory-protocols

We look forward to receiving your revised manuscript.

Kind regards,

Luca Paolo Ardigò, Ph.D.

Academic Editor

PLOS ONE

Additional Editor Comments (if provided):

Reviewer 1 still raises major issues: please, put all your effort to address them.

Reviewers' comments:

Reviewer's Responses to Questions

**Comments to the Author**

1. If the authors have adequately addressed your comments raised in a previous round of review and you feel that this manuscript is now acceptable for publication, you may indicate that here to bypass the “Comments to the Author” section, enter your conflict of interest statement in the “Confidential to Editor” section, and submit your "Accept" recommendation.

Reviewer #1: (No Response)

Reviewer #2: All comments have been addressed

Reviewer #3: All comments have been addressed

2. Is the manuscript technically sound, and do the data support the conclusions?

Reviewer #1: Partly

Reviewer #2: Yes

Reviewer #3: Yes

3. Has the statistical analysis been performed appropriately and rigorously? 

Reviewer #1: No

Reviewer #2: Yes

Reviewer #3: Yes

4. Have the authors made all data underlying the findings in their manuscript fully available?

Reviewer #1: Yes

Reviewer #2: Yes

Reviewer #3: Yes

5. Is the manuscript presented in an intelligible fashion and written in standard English?

Reviewer #1: Yes

Reviewer #2: Yes

Reviewer #3: Yes

6. Review Comments to the Author

Reviewer #1: Population stratification is still a big concern in this revision as commented before. The proportions of white vs black is not as similar as stated by the authors in the responses. In the paper it says “all ESP (white, 79%; black, 5%; black/white, 16%; and Asian, 1%) and a sub-sample (n = 59) of CON (white, 93%; black, 2%;mixed black/white, 3%; and Asian, 2%).” To me, 79% is pretty difference from 93%. Given that the majority of the samples are white, at least a subset analysis using only white is warranted.

Many hypotheses were tested in this paper, 10 SNPs x 5 outcome variables x 2 comparisons =100 already even before considering the two-way, three-way interactions. Adjusting for 10 tests is not enough. In addition, it is better to focus the paper on a smaller number of hypothesis. The sample size is limited in detection interaction effects, especially the three-way interaction.

Total genotype score based on SNPs that demonstrating differences is biased. Creating scores using genotypes from all 10 SNPs is not susceptible to such bias and it also reduces the number of comparisons.

The comment in the previous review: “Table 2. When there are only two categories of a variable only one category needs to be shown in the table since the second categories carries identical information. For example, for C vs G alleles or CC/GG vs GG genotypes of PPARA rs4253778, only one allele and one genotype group need to be shown.” What I meant is that if frequency of C allele is listed frequency of the G allele is just one minus that. But it is fine to leave both in. I would recommend putting actual counts in addition to frequencies in the table as well.

Reviewer #2: The revised manuscript is improved since the authors have addressed all the comments and concerns raised by the reviewers.

Reviewer #3: The authors have made good efforts to address the issues raised. The manuscript has scientific value and is suitable for publication.

7. PLOS authors have the option to publish the peer review history of their article (what does this mean?). If published, this will include your full peer review and any attached files.

Reviewer #1: No

Reviewer #2: Yes: Malcolm Collins

Reviewer #3: No

---

## [Author Response · Author response to Decision Letter 1]

24 Apr 2020

Dear Editor and Reviewers, 

Once again, we are very grateful to you all for taking the time to review our revised manuscript and for providing feedback for a second time, so thank you. 

Following your initial review of our original manuscript (submitted to PLoS ONE on 27th September 2019), we made significant changes in response to your welcomed comments, and we felt that we addressed all of them in a very comprehensive and fair manner. 

We are very pleased that Reviewer #2 and Reviewer #3 have confirmed their satisfaction with our changes and that they feel that our manuscript has scientific value and is suitable for publication. We have responded to Reviewer #1’s additional comments, below. 

As our manuscript will have been under review for seven months at the time of submitting our second revised manuscript, we would greatly appreciate it if this review could be expedited. Thank you for your understanding. 

Best wishes, 

Rob Erskine, on behalf of all co-authors

Reviewer #1: 

1. Population stratification is still a big concern in this revision as commented before. The proportions of white vs black is not as similar as stated by the authors in the responses. In the paper it says “all ESP (white, 79%; black, 5%; black/white, 16%; and Asian, 1%) and a sub-sample (n = 59) of CON (white, 93%; black, 2%;mixed black/white, 3%; and Asian, 2%).” To me, 79% is pretty difference from 93%. Given that the majority of the samples are white, at least a subset analysis using only white is warranted.

AUTHORS’ RESPONSE: As we stated in our original response to Reviewer #1’s comment on population stratification, we only have ethnic origin data in 38% of the control group. Therefore, we felt that making a comparison between Caucasian only ESP (n=422) and Caucasian only CON (n=54) would be unhelpful due to significantly reduced statistical power and the inevitability of making type II errors. Given the similarities between ESP and CON in terms of anthropometry, age and maturity status, and the similar percentages of various ethnic groups within ESP and the sub-sample of CON, we maintain that our genetic comparisons are still valid and our results compelling. However, in response to Reviewer #1’s request, we have investigated genotype/allele frequency distribution in participants of known Caucasian origin (ESP, n = 422; CON, n = 54), and found that ACTN3 XX genotype frequency distribution was higher (P = 0.004) in pre-PHV ESP (32.0%) compared to both CON (13.0%) and post-PHV ESP (20.4%), which supports our total cohort data. 

Consequently, we have included the following in the Discussion on P33 of the revised manuscript: 

“Nevertheless, despite ethnicity information only being available in 38% of CON, sub-sample analysis in participants of known Caucasian origin (ESP, n = 422; CON, n = 54) revealed that ACTN3 XX genotype frequency distribution was higher (P = 0.004) in pre-PHV ESP (32.0%) compared to both CON (13.0%) and post-PHV ESP (20.4%). This therefore supports our total cohort data.”

2. Many hypotheses were tested in this paper, 10 SNPs x 5 outcome variables x 2 comparisons =100 already even before considering the two-way, three-way interactions. Adjusting for 10 tests is not enough. In addition, it is better to focus the paper on a smaller number of hypothesis. The sample size is limited in detection interaction effects, especially the three-way interaction. 

AUTHORS’ RESPONSE: Each ANOVA model accounts for the multiple P values derived from the main effects and interaction effects within that model, i.e. main effect of genotype, 2-way interactions between genotype and maturation, 2-way interaction between genotype and athlete group, and three-way interaction between genotype, maturation and athlete group. 

It is also perfectly reasonable to perform a three-way ANOVA for each dependent variable (vertical CMJ, horizontal CMJ, agility, acceleration and sprint), as each of these variables are assessments of different aspects of physical performance. Thus, it is reasonable to hypothesise that some SNPs may be associated with just one or several of these performance variables, depending on the function of the SNP(s), which is not known in all of our 10 SNPs. 

We investigated the association 10 SNPs with these five performance variables. Therefore, we have performed 10 multiple comparisons within the same data set. This is why each of our FDR models (for the main effect and each of the interaction effects) included 10 P values. We feel that we have satisfactorily addressed this in our response to Reviewer #1’s initial comment regarding our statistical approach, i.e. as requested, we detailed our robust and rigorous statistical approach in our revised manuscript. 

3. Total genotype score based on SNPs that demonstrating differences is biased. Creating scores using genotypes from all 10 SNPs is not susceptible to such bias and it also reduces the number of comparisons. 

AUTHORS’ RESPONSE: We respectfully disagree with this comment. Firstly, the ‘performance’ TGS was calculated on the basis of a SNP being associated with any performance variable, and it could be associated solely with ESP or CON, or a particular maturation group within one or both groups. Therefore, we do not see how this TGS could be biased. Secondly, the ‘ESP’ TGS includes SNPs that had a higher/lower genotype/allele frequency distribution in one or multiple maturation groups in ESP vs. CON. We included only the associated SNPs, because they were the only ones of interest (we did not see any scientific justification for increasing the noise by including non-associated SNPs). Further, if a SNP is not associated with a dependent variable, we cannot be confident which genotype/allele is advantageous/disadvantageous. Referring to previous literature for this is not a viable solution, because ours is a completely novel population. Moreover, some of these SNPs have not been investigated in a sporting context before, making it impossible to know which is the advantageous genotype/allele. Finally, just because one SNP has a higher/lower genotype/allele frequency distribution does not mean that, when you include other SNPs in a combined TGS model, that the TGS will be higher in one cohort vs. another. This is because all ESP participants may not have all five beneficial genotypes. Indeed they did not but what was interesting (and the reason we performed the analysis this way) was that the ‘ESP’ TGS was higher in ESP than in CON, therefore providing evidence that a larger proportion of ESP had more of the beneficial genotypes than CON. Moreover (and another reason our TGS models were not biased), not all SNPs were associated with the same maturation group in ESP or CON but the TGS comparisons were total ESP TGS vs. total CON TGS. 

4. The comment in the previous review: “Table 2. When there are only two categories of a variable only one category needs to be shown in the table since the second categories carries identical information. For example, for C vs G alleles or CC/GG vs GG genotypes of PPARA rs4253778, only one allele and one genotype group need to be shown.” What I meant is that if frequency of C allele is listed frequency of the G allele is just one minus that. But it is fine to leave both in. I would recommend putting actual counts in addition to frequencies in the table as well. 

AUTHORS’ RESPONSE: We feel that including counts as well as percentages would clutter an already busy table. The sample sizes for each group are stated at the top of the table, and readers can calculate actual counts themselves using the percentages, if necessary. 

Reviewer #2: 

The revised manuscript is improved since the authors have addressed all the comments and concerns raised by the reviewers. 

AUTHORS’ RESPONSE: Thank you for confirming your satisfaction with our revised manuscript. We are very grateful to you for your time and feedback, which has helped improve our manuscript. 

Reviewer #3: 

The authors have made good efforts to address the issues raised. The manuscript has scientific value and is suitable for publication.

AUTHORS’ RESPONSE: Thank you also for confirming your satisfaction with the extensive changes we made to our manuscript following the first revision. We are very grateful for your time and feedback too, and we have no doubt that this has helped improve our manuscript.

---

## [Decision Letter · Decision Letter 2]

29 Apr 2020

PONE-D-19-26674R2

THE GENETIC PROFILE OF ELITE YOUTH SOCCER PLAYERS AND ITS ASSOCIATION WITH POWER AND SPEED DEPENDS ON MATURATION STATUS

PLOS ONE

Dear Dr Erskine,

Thank you for submitting your manuscript to PLOS ONE. After careful consideration, we feel that it has merit but does not fully meet PLOS ONE’s publication criteria as it currently stands. Therefore, we invite you to submit a revised version of the manuscript that addresses the points raised during the review process.

Please, try once more to address Reviewer 1's issues.

We would appreciate receiving your revised manuscript by Jun 13 2020 11:59PM. To enhance the reproducibility of your results, we recommend that if applicable you deposit your laboratory protocols in protocols.io, where a protocol can be assigned its own identifier (DOI) such that it can be cited independently in the future. For instructions see: http://journals.plos.org/plosone/s/submission-guidelines#loc-laboratory-protocols

We look forward to receiving your revised manuscript.

Kind regards,

Luca Paolo Ardigò, Ph.D.

Academic Editor

PLOS ONE

Additional Editor Comments (if provided):

Please, try once more to address Reviewer 1's issues.

Reviewers' comments:

Reviewer's Responses to Questions

**Comments to the Author**

1. If the authors have adequately addressed your comments raised in a previous round of review and you feel that this manuscript is now acceptable for publication, you may indicate that here to bypass the “Comments to the Author” section, enter your conflict of interest statement in the “Confidential to Editor” section, and submit your "Accept" recommendation.

Reviewer #1: (No Response)

2. Is the manuscript technically sound, and do the data support the conclusions?

Reviewer #1: (No Response)

3. Has the statistical analysis been performed appropriately and rigorously? 

Reviewer #1: (No Response)

4. Have the authors made all data underlying the findings in their manuscript fully available?

Reviewer #1: (No Response)

5. Is the manuscript presented in an intelligible fashion and written in standard English?

Reviewer #1: (No Response)

6. Review Comments to the Author

Reviewer #1: Regarding response to #1, not trying to be picky on words, however performing tests using Caucasians only does not “inevitably” introduce type II error since for type II error to occur the alternative hypothesis has to be true and we do not know if that is the case.

Only result of ACTN3 XX genotype is mentioned what about the other SNPs? It will be reassuring if the results of the Caucasian-analyses show similar trend, though not necessarily reaching statistical significance.

Regarding correction for multiple testing, I maintain my point of view that too many tests have been performed and the number of tests to be adjusted for should be more than 10. Having said that, the FDR adjusted p-values should be reported along with the unadjusted p-values so that the readers can weigh the evidence themselves. On line 247-248 “To control for multiple comparisons (10 SNPs and five performance variables), ….” , “and five performance variables” should be removed since they were not being controlled by the current FDR adjustment. Also a better place for this sentence is on line 241 after “main effects of genotypes”. This way it describes how overall association is evaluated before describing post-hoc comparison.

Regarding bias in “TGS”, even though the SNPs were selected based on different performance outcomes it is conceivable that these outcomes are also associated with ESP and CON group status. Therefore this preselection process can be biased.

7. PLOS authors have the option to publish the peer review history of their article (what does this mean?). If published, this will include your full peer review and any attached files.

Reviewer #1: No

---

## [Author Response · Author response to Decision Letter 2]

29 Apr 2020

Dear Editor, 

We have responded to Reviewer #1’s additional comments, below. 

As our manuscript will have been under review for over seven months at the time of submitting the fourth version of our manuscript, we would greatly appreciate it if this final review could be expedited. Thank you for your understanding. 

Best wishes, 

Rob Erskine, on behalf of all co-authors

Reviewer #1: 

1. Regarding response to #1, not trying to be picky on words, however performing tests using Caucasians only does not “inevitably” introduce type II error since for type II error to occur the alternative hypothesis has to be true and we do not know if that is the case. Only result of ACTN3 XX genotype is mentioned what about the other SNPs? It will be reassuring if the results of the Caucasian-analyses show similar trend, though not necessarily reaching statistical significance.

AUTHORS’ RESPONSE: If the sample size is small, it increases the chance of making type II errors. Only using 38% of the CON group (not because they were the only Caucasian CON participants but because they were the only ones we had ethnicity information for) severely reduces the sample size, and therefore increases the chance of making type II errors. We were willing to report the ACTN3 genotype frequency data in Caucasian only participants in response to Reviewer #1’s request, because it did not confuse the main message of the manuscript. Due to only including 38% CON, the other genotype/allele data showed a similar trend to the overall genotype/allele data but did not reach statistical significance (PPARA GG: post-PHV ESP 59.2%, pre-PHV ESP 68.9%, CON 72.2%, P = 0.069; PPARA G-allele: post-PHV ESP 77.2%, pre-PHV ESP 83.0%, CON 85.2%, P = 0.060; AGT AA: post-PHV ESP 24.6%, pre-PHV ESP 35.0%, CON 27.8%, P = 0.133; AGT A-allele: post-PHV ESP 49.3%, pre-PHV ESP 56.8%, CON 56.5%, P = 0.109; and NOS3 T-allele: ESP 63.5%, CON 57.4%, P = 0.217). However, including those under-powered data would confuse the manuscript considerably but we trust this similar trend satisfies the reviewer. We believe including the Caucasian only ACTN3 genotype frequency information in the revised manuscript is a fair compromise. 

2. Regarding correction for multiple testing, I maintain my point of view that too many tests have been performed and the number of tests to be adjusted for should be more than 10. Having said that, the FDR adjusted p-values should be reported along with the unadjusted p-values so that the readers can weigh the evidence themselves. On line 247-248 “To control for multiple comparisons (10 SNPs and five performance variables), ….” , “and five performance variables” should be removed since they were not being controlled by the current FDR adjustment. Also a better place for this sentence is on line 241 after “main effects of genotypes”. This way it describes how overall association is evaluated before describing post-hoc comparison. 

AUTHORS’ RESPONSE: As explained in our initial Response to reviewers’ comments document, and in the first revised manuscript, the original (“unadjusted”) P values were compared with the respective B-H critical values (line 225-229). Reporting “adjusted” B-H P values can be confusing in itself but adding this information to the original (“unadjusted”) P values would complicate and confuse the results section, and the overall manuscript. Comparing the original (unadjusted) P values with their respective B-H critical values is normal practice and we see no reason to change this. However, as per Reviewer #1’s request. we have removed “and five variables” from line 247 and moved this sentence to line 241. 

3. Regarding bias in “TGS”, even though the SNPs were selected based on different performance outcomes it is conceivable that these outcomes are also associated with ESP and CON group status. Therefore this preselection process can be biased.

 AUTHORS’ RESPONSE: For a full rebuttal to this comment, please refer to our previous Response to reviewers’ comments document. In brief, the ‘performance’ TGS comprised SNPs that were associated with different performance variables in CON and/or ESP at different stages of maturity. Therefore, we maintain there was no bias.

---

## [Decision Letter · Decision Letter 3]

28 May 2020

THE GENETIC PROFILE OF ELITE YOUTH SOCCER PLAYERS AND ITS ASSOCIATION WITH POWER AND SPEED DEPENDS ON MATURITY STATUS

PONE-D-19-26674R3

Dear Dr. Erskine,

We are pleased to inform you that your manuscript has been judged scientifically suitable for publication and will be formally accepted for publication once it complies with all outstanding technical requirements.

With kind regards,

Luca Paolo Ardigò, Ph.D.

Academic Editor

PLOS ONE

Additional Editor Comments (optional):

In my opinion, manuscript deserves publication in spite of not unanimous acceptance recommendation by all three expert reviewers. Congratulations to authors.

Reviewers' comments:

Reviewer's Responses to Questions

**Comments to the Author**

1. If the authors have adequately addressed your comments raised in a previous round of review and you feel that this manuscript is now acceptable for publication, you may indicate that here to bypass the “Comments to the Author” section, enter your conflict of interest statement in the “Confidential to Editor” section, and submit your "Accept" recommendation.

Reviewer #1: (No Response)

2. Is the manuscript technically sound, and do the data support the conclusions?

Reviewer #1: (No Response)

3. Has the statistical analysis been performed appropriately and rigorously? 

Reviewer #1: (No Response)

4. Have the authors made all data underlying the findings in their manuscript fully available?

Reviewer #1: (No Response)

5. Is the manuscript presented in an intelligible fashion and written in standard English?

Reviewer #1: (No Response)

6. Review Comments to the Author

Reviewer #1: (No Response)

7. PLOS authors have the option to publish the peer review history of their article (what does this mean?). If published, this will include your full peer review and any attached files.

Reviewer #1: No

---

## [Editor Report · Acceptance letter]

4 Jun 2020

PONE-D-19-26674R3 

THE GENETIC PROFILE OF ELITE YOUTH SOCCER PLAYERS AND ITS ASSOCIATION WITH POWER AND SPEED DEPENDS ON MATURITY STATUS 

Dear Dr. Erskine:

I'm pleased to inform you that your manuscript has been deemed suitable for publication in PLOS ONE. Congratulations! Your manuscript is now with our production department. 

Kind regards, 

on behalf of

Dr. Luca Paolo Ardigò 

Academic Editor

PLOS ONE